# Reward-agnostic Fine-tuning: Provable Statistical Benefits of Hybrid Reinforcement Learning

**Gen Li**[*]  **Wenhao Zhan**[*]  **Jason D. Lee**  **Yuejie Chi**  **Yuxin Chen**[†]
CUHK           Princeton          Princeton         CMU            UPenn

## Abstract

This paper studies tabular reinforcement learning (RL) in the hybrid setting, which assumes access to both an offline dataset and online interactions with the unknown environment. A central question boils down to how to efficiently utilize online data to strengthen and complement the offline dataset and enable effective policy fine-tuning. Leveraging recent advances in reward-agnostic exploration and offline RL, we design a three-stage hybrid RL algorithm that beats the best of both worlds — pure offline RL and pure online RL — in terms of sample complexities. The proposed algorithm does not require any reward information during data collection. Our theory is developed based on a new notion called *single-policy partial concentrability*, which captures the trade-off between distribution mismatch and miscoverage and guides the interplay between offline and online data.

## 1 Introduction

As reinforcement learning (RL) shows promise in achieving super-human empirical success across diverse fields (e.g., games (Silver et al., 2016; Vinyals et al., 2019; Berner et al., 2019; Mnih et al., 2013), robotics (Brambilla et al., 2013), autonomous driving (Shalev-Shwartz et al., 2016)), theoretical understanding about RL has also been substantially expanded, with the aim of distilling fundamental principles that can inform and guide practice. Among all sorts of theoretical questions being pursued, how to make the best use of data emerges as a question of profound interest for problems with enormous dimensionality.

There are at least two mainstream mechanisms when it comes to data collection: online RL and offline RL. Let us briefly describe their attributes and differences as follows.

*Online RL.* In this setting, an agent learns how to maximize her cumulative reward through interaction with the unknown environment (by, say, executing a sequence of adaptively chosen actions and utilizing the instantaneous feedback of the environment). Given that all information about the environment is obtained through real-time data collection, the main challenge lies in how to (optimally) manage the trade-off between exploration and exploitation. Towards this, one popular approach advertises the principle of optimism in the face of uncertainty — e.g., employing upper confidence bounds during value estimation to guide exploration — whose effectiveness has been shown for both the tabular case (Auer and Ortner, 2006; Jaksch et al., 2010; Azar et al., 2017; Dann et al., 2017; Jin et al., 2018; Bai et al., 2019; Dong et al., 2019; Zhang et al., 2020b; Ménard et al., 2021b; Li et al., 2021b) and the case with function approximation (Jin et al., 2020b; Zanette et al., 2020; Zhou et al., 2021a; Li et al., 2021a; Du et al., 2021; Jin et al., 2021a; Foster et al., 2021; Chen et al., 2022b).

*Offline RL.* In contrast, offline RL assumes access to a pre-collected dataset, without given permission to perform any further data collection. The feasibility of reliable offline RL depends heavily

---

[*]The first two authors contributed equally.

[†]Corresponding author

on the quality of the dataset at hand. A central challenge stems from the presence of distribution shift: the distribution of the offline dataset might differ significantly from that induced by the target policy. Another common challenge arises from insufficient data coverage: a nontrivial fraction of the state-action pairs might be inadequately visited in the available dataset, thus precluding one from faithfully evaluating many policies based solely on the offline dataset. To circumvent these obstacles, recent works proposed the principle of pessimism in the face of uncertainty, recommending caution when selecting poorly visited actions (Liu et al., 2020; Kumar et al., 2020; Jin et al., 2021b; Rashidinejad et al., 2021; Uehara and Sun, 2021; Li et al., 2022; Yin et al., 2021; Shi et al., 2022; Cai et al., 2022). Without requiring uniform coverage of all policies, the pessimism approach proves effective as long as the so-called *single-policy concentrability* is satisfied, which only assumes adequate coverage over the part of the state-action space reachable by the desirable policy.

In reality, however, both mechanisms above come with limitations. For instance, even the single-policy concentrability requirement might be too stringent (and hence fragile) for offline RL, as it is not uncommon for the historical dataset to miss a small yet essential part of the state-action space. Pure online RL might also be overly restrictive, given that there might be information from past data that could help initialize online exploration and mitigate the burden of further data collection.

All this motivates the studies of hybrid RL, a scenario where the agent has access to an offline dataset while, in the meantime, (limited) online data collection is permitted as well. Oftentimes, this scenario is practically not only feasible but also appealing: on the one hand, offline data provides useful information for policy pre-training, while further online exploration helps enrich existing data and allows for effective policy fine-tuning. As a matter of fact, multiple empirical works (Rajeswaran et al., 2017; Vecerik et al., 2017; Kalashnikov et al., 2018; Hester et al., 2018; Nair et al., 2018, 2020) indicated that combining online RL with offline datasets outperforms both pure online RL and pure offline RL. Nevertheless, theoretical pursuits about hybrid RL are lagging behind. Two recent works Ross and Bagnell (2012); Xie et al. (2021b) studied a restricted setting, where the agent is aware of a Markovian behavior policy (a policy that generates offline data) and can either execute the behavior policy or any other adaptive choice to draw samples in each episode; in this case, Xie et al. (2021b) proved that under the single-policy concentrability assumption of the offline dataset, having perfect knowledge about the behavior policy does not improve online exploration in the minimax sense. Another strand of works Song et al. (2022); Nakamoto et al. (2023); Wagenmaker and Pacchiano (2022) looked at a more general offline dataset and investigated how to leverage offline data in online exploration. From the sample complexity viewpoint, Wagenmaker and Pacchiano (2022) studied the statistical benefits of hybrid RL in the presence of linear function approximation; the result therein, however, required strong assumptions on data coverage (i.e., all-policy concentrability) and fell short of unveiling provable gains in the tabular case (as we shall elucidate momentarily). In light of such theoretical inadequacy in previous works, this paper is motivated to pursue the following question:

> *Does hybrid RL allow for improved sample complexity compared to pure online or offline RL in the tabular case?*

## 1.1 Main contributions

We deliver an affirmative answer to the above question. Further relaxing the single-policy concentrability assumption, we introduce a relaxed notation called single-policy *partial* concentrability (to be made precise in Definition 2), which (i) allows the dataset to miss a fraction of the state-action space visited by the optimal policy and (ii) captures the tradeoff between distribution mismatch and lack of coverage. Armed with this notion, our results reveal provable statistical benefits of hybrid RL compared with both pure online and offline RL. The main contributions are summarized below.

*A novel three-stage algorithm.* We design a new hybrid RL algorithm consisting of three stages. In the first stage, we obtain crude estimation of the occupancy distribution $d^\pi$ w.r.t. any policy $\pi$ as well as the data distribution $d^{\text{off}}$ of the offline dataset. The second stage performs online exploration; in particular, we execute one exploration policy to imitate the offline dataset and another one to explore the inadequately visited part of the unknown environment, with both policies computed by approximately solving convex optimization sub-problems. Notably, these two stages do not count on the availability of reward information, and thus operate in a reward-agnostic manner. The final stage then invokes the state-of-the-art offline RL algorithm for policy learning, on the basis of all data we have available (including both online and offline data).

*Computationally efficient subroutines.* Throughout the first two stages of the algorithm, we need to solve a couple of convex sub-problems with exponentially large dimensions. In order to attain computational efficiency, we design efficient Frank-Wolfe-type paradigms to solve the sub-problems approximately, which run in polynomial time. This plays a crucial role in ensuring computational tractability of the proposed three-stage algorithm.

*Improved sample complexity.* We characterize the sample complexity of our algorithm (see Theorem 1), which provably improves upon both pure online and offline RL. On the one hand, hybrid RL achieves strictly enhanced performance compared to pure offline RL (assuming the same sample size) when the offline data falls short of covering all state-action pairs reachable by the desired policy. On the other hand, the sample size allocated to online exploration in our algorithm might only need to be proportional to the fraction $\sigma$ of the state-action space uncovered by the offline dataset, thus resulting in sample size saving in general compared to pure online RL (a case with $\sigma = 1$).

**Notation.** Let us also introduce several useful notation. For integer $m > 0$, we let $[m]$ represent the set $\{1, \cdots, m\}$. For any set $\mathcal{B}$, we denote by $\mathcal{B}^c$ its complement. For any policy $\pi_0$, we let $\mathbb{1}_{\pi_0} : \Pi \to \{0, 1\}$ be an indicator function such that $\mathbb{1}_{\pi_0}(\pi) = 1$ if $\pi = \pi_0$ and $\mathbb{1}_{\pi_0}(\pi) = 0$ otherwise. For any finite set $\mathcal{A}$, we denote by $\Delta(\mathcal{A})$ the probability simplex over $\mathcal{A}$. Letting $\mathcal{X} := \left(S, A, H, \frac{1}{\varepsilon}, \frac{1}{\delta}\right)$, we use the notation $f(\mathcal{X}) = O(g(\mathcal{X}))$ or $f(\mathcal{X}) \lesssim g(\mathcal{X})$ to indicate the existence of a universal constant $C_1 > 0$ such that $f \leq C_1 g$, the notation $f(\mathcal{X}) \gtrsim g(\mathcal{X})$ to indicate that $g(\mathcal{X}) = O(f(\mathcal{X}))$, and the notation $f(\mathcal{X}) \asymp g(\mathcal{X})$ to mean that $f(\mathcal{X}) \lesssim g(\mathcal{X})$ and $f(\mathcal{X}) \gtrsim g(\mathcal{X})$ hold simultaneously. The notation $\widetilde{O}(\cdot)$ is defined in the same way as $O(\cdot)$ except that it hides logarithmic factors.

## 2 Preliminaries and problem settings

**Episodic finite-horizon MDPs.** We study episodic finite-horizon Markov decision processes with $S$ states, $A$ actions, and horizon length $H$. We use $\mathcal{M} = (\mathcal{S}, \mathcal{A}, H, P = \{P_h\}_{h=1}^H, r = \{r_h\}_{h=1}^H)$ to represent such an MDP, where $\mathcal{S} = [S]$ and $\mathcal{A} = [A]$ represent the state space and the action space, respectively. For each step $h \in [H]$, we let $P_h : \mathcal{S} \times \mathcal{A} \to \Delta(\mathcal{S})$ represent the transition probability at this step, such that taking action $a$ in state $s$ at step $h$ yields a transition to the next state drawn from the distribution $P_h(\cdot \,|\, s, a)$; throughout the paper, we often employ the shorthand notation $P_{h,s,a} := P_h(\cdot|s, a)$. Another ingredient is the reward function specified by $r_h : \mathcal{S} \times \mathcal{A} \to [0, 1]$ at step $h$; namely, the agent will receive an immediate reward $r_h(s, a)$ upon executing action $a$ in state $s$ at step $h$. It is assumed that the reward function is fully revealed upon completion of online data collection. Additionally, we assume throughout that each episode of the MDP starts from an initial state independently generated from some (unknown) initial state distribution $\rho \in \Delta(\mathcal{S})$.

A time-inhomogeneous Markovian policy is often denoted by $\pi = \{\pi_h\}_{h=1}^H$ with $\pi_h : \mathcal{S} \to \Delta(\mathcal{A})$, where $\pi_h(\cdot \,|\, s)$ characterizes the (randomized) action selection probability of the agent in state $s$ at step $h$. If $\pi$ is a deterministic policy, then we often abuse the noation and let $\pi_h(s)$ represent the action selected in state $s$ at step $h$. We find it convenient to introduce the following notation:

$$\Pi := \text{the set of all deterministic policies.} \tag{1}$$

We also need to handle mixed deterministic policies (i.e., each realization of the policy is randomly drawn from a mixture of deterministic policies). A mixed deterministic policy $\pi^{\mathsf{mixed}}$ is denoted by

$$\pi^{\mathsf{mixed}} = \sum_{\pi \in \Pi} \mu(\pi)\pi = \mathbb{E}_{\pi \sim \mu}[\pi] \qquad \text{for some } \mu \in \Delta(\Pi). \tag{2}$$

Moreover, for any policy $\pi$, we define its associated value function (resp. Q-function) as follows, representing the expected cumulative rewards conditioned on an initial state (resp. an initial state-action pair):

$$V_h^\pi(s) := \mathbb{E}_\pi\left[\sum_{h':h \leq h' \leq H} r_{h'}(s, a) \,\middle|\, s_h = s\right], \qquad \forall s \in \mathcal{S};$$

$$Q_h^\pi(s, a) := \mathbb{E}_\pi\left[\sum_{h':h \leq h' \leq H} r_{h'}(s, a) \,\middle|\, s_h = s, a_h = a\right], \qquad \forall (s, a) \in \mathcal{S} \times \mathcal{A}.$$

Here, the expectation is over the length-$H$ sample trajectory $(s_1, a_1, s_2, a_2, \ldots, s_H, a_H)$ when executing policy $\pi$ in $\mathcal{M}$, where $s_h$ (resp. $a_h$) denotes the state (resp. action) at step $h$ of this trajectory.

When the initial state is drawn from $\rho$, we further augment the notation and denote

$$V_1^\pi(\rho) = \mathbb{E}_{s \sim \rho}\big[V_1^\pi(s)\big].$$

Importantly, there exists at least one deterministic policy, denoted by $\pi^\star$ throughout, that is able to maximize $V_h^\pi(s)$ and $Q_h^\pi(s,a)$ simultaneously for all $(h,s,a) \in [H] \times \mathcal{S} \times \mathcal{A}$; namely,

$$V_h^\star(s) \coloneqq V_h^{\pi^\star}(s) = \max_\pi V_h^\pi(s), \qquad Q_h^\star(s,a) \coloneqq Q_h^{\pi^\star}(s,a) = \max_\pi Q_h^\pi(s,a), \quad \forall (s,a) \in \mathcal{S} \times \mathcal{A}.$$

Moving beyond value functions and Q-functions, we would like to define, for each policy $\pi$, the associated state-action occupancy distribution $d^\pi = [d_h^\pi]_{1 \le h \le H}$ such that

$$d_h^\pi(s,a) \coloneqq \mathbb{P}(s_h = s, a_h = a \,|\, \pi), \qquad \forall (s,a,h) \in \mathcal{S} \times \mathcal{A} \times [H];$$

in other words, this is the probability of the state-action pair $(s,a) \in \mathcal{S} \times \mathcal{A}$ being visited by $\pi$ at step $h$. We shall also overload $d^\pi$ to represent the state occupancy distribution such that

$$d_h^\pi(s) \coloneqq \sum\nolimits_{a \in \mathcal{A}} d_h^\pi(s,a) = \mathbb{P}(s_h = s \,|\, \pi), \qquad \forall (s,h) \in \mathcal{S} \times [H]. \tag{3}$$

Given that each episode always starts with a state drawn from $\rho$, it is easily seen that $d_1^\pi(s) = \rho(s)$ for any policy $\pi$ and any $s \in \mathcal{S}$.

**Sampling mechanism.** We consider a hybrid RL setting that assumes access to a historical dataset as well as the ability to further explore the environment via real-time sampling, as detailed below.

*Offline data.* Suppose that we have available a historical dataset (also called an offline dataset)

$$\mathcal{D}^{\mathsf{off}} = \big\{\tau^{k,\mathsf{off}}\big\}_{1 \le k \le K^{\mathsf{off}}}, \tag{4}$$

containing $K^{\mathsf{off}}$ sample trajectories each of length $H$. Here, the $k$-th trajectory in $\mathcal{D}^{\mathsf{off}}$ is denoted by

$$\tau^{k,\mathsf{off}} = \big(s_1^{k,\mathsf{off}}, a_1^{k,\mathsf{off}}, \ldots, s_H^{k,\mathsf{off}}, a_H^{k,\mathsf{off}}\big), \tag{5}$$

where $s_h^{k,\mathsf{off}}$ and $a_h^{k,\mathsf{off}}$ indicate respectively the state and action at step $h$ of this trajectory $\tau^{k,\mathsf{off}}$. It is assumed that each trajectory $\tau^{k,\mathsf{off}}$ is drawn *independently* using policy $\pi^{\mathsf{off}}$, which takes the form of a mixture of deterministic policies

$$\pi^{\mathsf{off}} = \mathbb{E}_{\pi \sim \mu^{\mathsf{off}}}\big[\pi\big] \qquad \text{with } \mu^{\mathsf{off}} \in \Delta(\Pi). \tag{6}$$

Note that the learner only has access to the data samples but not $\pi^{\mathsf{off}}$. Throughout the paper, we use $d^{\mathsf{off}} = \{d_h^{\mathsf{off}}\}_{1 \le h \le H}$ to represent the occupancy distribution of this offline dataset such that

$$d_h^{\mathsf{off}}(s,a) \coloneqq \mathbb{P}\big((s_h^{k,\mathsf{off}}, a_h^{k,\mathsf{off}}) = (s,a)\big), \qquad \forall (s,a,h) \in \mathcal{S} \times \mathcal{A} \times [H]. \tag{7}$$

*Online exploration.* In addition to the offline dataset, the learner is allowed to interact with the unknown environment and collect more data in real time, in the hope of compensating for the insufficiency of the pre-collected data at hand and fine-tuning the policy estimate. More specifically, the learner is able to sample $K^{\mathsf{on}}$ trajectories sequentially. In each sample trajectory,

- the initial state is generated independently from an (unknown) distribution $\rho \in \Delta(\mathcal{S})$;
- the learner selects a policy to execute the MDP, obtaining a sample trajectory of length $H$.

The total number of sample trajectories is thus given by

$$K = K^{\mathsf{off}} + K^{\mathsf{on}}. \tag{8}$$

**Concentrability assumptions for the offline dataset.** To quantify the quality of the historical dataset, prior offline RL literature introduced the following single-policy concentrability coefficient based on certain density ratio of interest; see, e.g., Rashidinejad et al. (2021); Li et al. (2022).

**Definition 1** (Single-policy concentrability). *The single-policy concentrability coefficient $C^\star$ of the offline dataset $\mathcal{D}^{\mathsf{off}}$ is defined as*

$$C^\star \coloneqq \max_{(s,a,h) \in \mathcal{S} \times \mathcal{A} \times [H]} \frac{d_h^{\pi^\star}(s,a)}{d_h^{\mathsf{off}}(s,a)}. \tag{9}$$

In words, $C^\star$ employs the $\ell_\infty$-norm of the density ratio $d^{\pi^\star}/d^{\mathsf{off}}$ to capture the shift of distributions between the occupancy distribution induced by the desired policy $\pi^\star$ and the data distribution at hand. The terminology "single-policy" underscores that Definition 1 only compares the offline data distribution against the one generated by a single policy $\pi^\star$, which stands in stark contrast to other all-policy concentrability coefficients that are defined to account for all policies simultaneously.

One notable fact about Definition 1 is that: for $C^\star$ to be finite, the historical data distribution needs to cover all state-action-step tuples reachable by $\pi^\star$. This requirement is, in general, inevitable if only the offline dataset is available; see the minimax lower bounds in Rashidinejad et al. (2021); Li et al. (2022) for more precise justifications. However, a requirement of this kind could be overly stringent for the hybrid setting considered herein, as the issue of incomplete coverage can potentially be overcome with the aid of online data collection. In light of this observation, we generalize Definition 1 to account for the trade-offs between distributional mismatch and partial coverage.

**Definition 2** (Single-policy partial concentrability). *For any $\sigma \in [0,1]$, the single-policy partial concentrability coefficient $C^\star(\sigma)$ of the offline dataset $\mathcal{D}^{\mathsf{off}}$ is defined as*

$$C^\star(\sigma) \coloneqq \min\left\{ \max_{1 \le h \le H} \max_{(s,a) \in \mathcal{G}_h} \frac{d_h^{\pi^\star}(s,a)}{d_h^{\mathsf{off}}(s,a)} \;\middle|\; \{\mathcal{G}_h\}_{1 \le h \le H} \in \mathcal{G}(\sigma) \right\}, \tag{10}$$

*where*

$$\mathcal{G}(\sigma) \coloneqq \left\{ \{\mathcal{G}_h\}_{1 \le h \le H} \subseteq \mathcal{S} \times \mathcal{A} \;\middle|\; \frac{1}{H}\sum_{h=1}^{H}\sum_{(s,a) \notin \mathcal{G}_h} d_h^{\pi^\star}(s,a) \le \sigma \right\}. \tag{11}$$

In Definition 2, we allow a fraction of the state-action space reachable by $\pi^\star$ to be insufficiently covered (as reflected in the definition of $\mathcal{G}(\sigma)$ measured by the state-action occupancy distribution) — hence the terminology "partial". Intuitively, $\mathcal{G}_h$ corresponds to a set of state-action pairs that undergo reasonable distribution shift (so that the corresponding density ratio does not rise above $C^\star(\sigma)$), whereas the total occupancy density of its complement subset $\mathcal{G}_h^c$ induced by $\pi^\star$ is under control (i.e., no larger than $\sigma$ when averaged across steps). As a self-evident fact, $C^\star(\sigma)$ is non-increasing in $\sigma$; this means that as $\sigma$ increases, we might incur a less severe distribution shift in a restricted part, at the price of less coverage. In this sense, $C^\star(\sigma)$ reflects certain tradeoffs between distribution shift and coverage. Clearly, $C^\star(\sigma)$ reduces to $C^\star$ in Definition 1 by taking $\sigma = 0$.

**Goal.** Given a historical dataset $\mathcal{D}^{\mathsf{off}}$ containing $K^{\mathsf{off}}$ sample trajectories, we would like to design an online exploration scheme, in conjunction with the accompanying policy learning algorithm, so as to achieve desirable policy learning (or policy fine-tuning) in a data-efficient manner. Ideally, we would expect a hybrid RL algorithm to harvest provable statistical benefits compared to both purely online RL and purely offline RL approaches.

## 3 Algorithm

In this section, we propose a new algorithm to tackle the hybrid RL setting. Our algorithm design leverages recent ideas developed in offline RL and reward-agnostic exploration to improve sample efficiency. Our algorithm consists of three stages to be described shortly; informally, the first two stages conduct reward-agnostic exploration to imitate and complement the offline dataset, whereas the third stage invokes a sample-optimal offline RL algorithm to compute a near-optimal policy.

In the sequel, we split the offline dataset $\mathcal{D}^{\mathsf{off}}$ into two halves:

$$\mathcal{D}^{\mathsf{off},1} \qquad \text{and} \qquad \mathcal{D}^{\mathsf{off},2}, \tag{12}$$

where $\mathcal{D}^{\mathsf{off},1}$ (resp. $\mathcal{D}^{\mathsf{off},2}$) consists of the first (resp. last) $K^{\mathsf{off}}/2$ independent trajectories from $\mathcal{D}^{\mathsf{off}}$. As we shall also see momentarily, online exploration in the proposed algorithm — which collects $K^{\mathsf{on}}$ trajectories in total — can be divided into three parts, collecting $K^{\mathsf{on}}_{\mathsf{prepare}}$, $K^{\mathsf{on}}_{\mathsf{imitate}}$ and $K^{\mathsf{on}}_{\mathsf{explore}}$ sample trajectories, respectively. Throughout this paper, for simplicity we choose

$$K^{\mathsf{on}}_{\mathsf{prepare}} = K^{\mathsf{on}}_{\mathsf{imitate}} = K^{\mathsf{on}}_{\mathsf{explore}} = K^{\mathsf{on}}/3. \tag{13}$$

### 3.1 A three-stage algorithm

We now elaborate on the three stages of the proposed algorithm. Due to space limitation, the pseudocode of the complete algorithm is provided in Appendix B.

**Stage 1: estimation of the occupancy distributions.** As a preparatory step for reward-agnostic exploration, we first attempt to estimate the occupancy distribution induced by any policy as well as the occupancy distribution $d^{\mathsf{off}}$ associated with the historical dataset, as described below.

*Estimating $d^{\pi}$ for any policy $\pi$.* In this step, we would like to sample the environment and collect a set of sample trajectories, in a way that allows for reasonable estimation of the occupancy distribution $d^{\pi}$ induced by any policy $\pi$. For this purpose, we invoke the exploration strategy and the accompanying estimation scheme developed in Li et al. (2023). Working forward (i.e., from $h = 1$ to $H$), this approach collects, for each step $h$, a set of $N$ sample trajectories in order to facilitate estimation of the occupancy distributions, which amounts to a total number of

$$NH =: K^{\mathsf{on}}_{\mathsf{prepare}} = K^{\mathsf{on}}/3 \tag{14}$$

sample trajectories collected in this stage. See Algorithm 3 in Appendix D.1 for a precise description of this strategy. Noteworthily, while Algorithm 3 specifies how to estimate $\widehat{d^{\pi}}$ for any policy $\pi$, we won't need to compute it explicitly unless this policy $\pi$ is encountered during the subsequent steps of the algorithm; in other words, $\widehat{d^{\pi}}$ should be viewed as a sort of "function handle" that will only be executed when called later.

*Estimating $d^{\mathsf{off}}$ for the historical dataset $\mathcal{D}^{\mathsf{off}}$.* In addition, we are in need of estimating the occupancy distribution $d^{\mathsf{off}}$. Towards this end, we propose the following empirical estimate using the $K^{\mathsf{off}}/2$ sample trajectories from $\mathcal{D}^{\mathsf{off},1}$:

$$\widehat{d}^{\mathsf{off}}_h(s,a) = \frac{2N^{\mathsf{off}}_h(s,a)}{K^{\mathsf{off}}} \mathbb{1}\left( \frac{N^{\mathsf{off}}_h(s,a)}{K^{\mathsf{off}}} \geq c_{\mathsf{off}} \left\{ \frac{\log \frac{HSA}{\delta}}{K^{\mathsf{off}}} + \frac{H^4 S^4 A^4 \log \frac{HSA}{\delta}}{N} + \frac{SA}{K^{\mathsf{on}}} \right\} \right) \tag{15}$$

for all $(s,a) \in \mathcal{S} \times \mathcal{A}$, where $c_{\mathsf{off}} > 0$ is some universal constant. Here, $1 - \delta$ indicates the target success probability, and

$$N^{\mathsf{off}}_h(s,a) = \sum_{k=1}^{K^{\mathsf{off}}/2} \mathbb{1}\left( s^{k,\mathsf{off}}_h = s, a^{k,\mathsf{off}}_h = a \right), \qquad \forall (s,a) \in \mathcal{S} \times \mathcal{A}. \tag{16}$$

In other words, $\widehat{d}^{\mathsf{off}}_h(s,a)$ is taken to be the empirical visitation frequency of $(s,a)$ in $\mathcal{D}^{\mathsf{off},1}$ if $(s,a)$ is adequately visited, and zero otherwise. The cutoff threshold will be made clear in our analysis.

**Stage 2: online exploration.** Armed with the above estimates of the occupancy distributions, we can readily proceed to compute the desired exploration policies and sample the environment. We seek to devise two exploration strategies, with one strategy selected to imitate the offline dataset, and the other one employed to explore the insufficiently visited territory. As a preliminary fact, if we have a dataset containing $K$ independent trajectories — generated independently from a mixture of deterministic policies with occupancy distribution $d^{\mathsf{b}}$ — then it has been shown previously (see, e.g., Li et al. (2023, Section 3.3)) that the model-based offline approach is able to compute a policy $\widehat{\pi}$ obeying

$$V^{\star}(\rho) - V^{\widehat{\pi}}(\rho) \lesssim H \left[ \sum_h \sum_{s,a} \frac{d^{\pi^{\star}}_h(s,a)}{1/H + K^{\mathsf{on}} d^{\mathsf{b}}_h(s,a)} \right]^{\frac{1}{2}}. \tag{17}$$

This upper bound in (17) provides a guideline regarding how to design a sample-efficient exploration scheme.

*Imitating the offline dataset.* The offline dataset $\mathcal{D}^{\mathsf{off}}$ is most informative when it contains expert data, a scenario when the data distribution resembles the distribution induced by the optimal policy $\pi^{\star}$. If this is the case, then it is desirable to find a policy similar to $\pi^{\mathsf{off}}$ in (6) (the mixed policy generating $\mathcal{D}^{\mathsf{off}}$) and employ it to collect new data, in order to retain and further strength the benefits of

such offline data. To do so, we attempt to approximate $d^{\pi^\star}$ by $\widehat{d}^{\mathsf{off}}$ in (17) when attempting to minimize (17). In fact, we would like to compute a mixture of deterministic policies by (approximately) solving the following optimization problem:

$$\mu^{\mathsf{imitate}} \approx \arg \min_{\mu \in \Delta(\Pi)} \sum_{h=1}^{H} \sum_{s \in \mathcal{S}} \max_{a \in \mathcal{A}} \frac{\widehat{d}_h^{\mathsf{off}}(s,a)}{\frac{1}{K^{\mathsf{on}}H} + \mathbb{E}_{\pi' \sim \mu}\big[\widehat{d}_h^{\pi'}(s,a)\big]}, \tag{18}$$

which is clearly equivalent to

$$\mu^{\mathsf{imitate}} \approx \arg \min_{\mu \in \Delta(\Pi)} \max_{\pi:\mathcal{S} \times [H] \to \Delta(\mathcal{A})} \sum_{h=1}^{H} \sum_{s \in \mathcal{S}} \mathbb{E}_{a \sim \pi_h(\cdot|s)} \left[ \frac{\widehat{d}_h^{\mathsf{off}}(s,a)}{\frac{1}{K^{\mathsf{on}}H} + \mathbb{E}_{\pi' \sim \mu}\big[\widehat{d}_h^{\pi'}(s,a)\big]} \right]. \tag{19}$$

In order to solve this minimax problem (19) (note that its objective function is convex in $\mu$), we resort to the Follow-The-Regularized-Leader (FTRL) strategy from the online learning literature (Shalev-Shwartz, 2012); more specifically, we perform the following updates iteratively for $t = 1, \ldots, T_{\mathsf{max}}$:

$$\pi_h^{t+1}(\cdot \mid s) \propto \exp\left( \eta \sum_{k=1}^{t} \frac{\widehat{d}_h^{\mathsf{off}}(s,\cdot)}{\frac{1}{K^{\mathsf{on}}H} + \mathbb{E}_{\pi' \sim \mu^k}\big[\widehat{d}_h^{\pi'}(s,\cdot)\big]} \right), \qquad \forall s \in \mathcal{S}, \tag{20a}$$

$$\mu^{t+1} \approx \arg \min_{\mu \in \Delta(\Pi)} \sum_{h=1}^{H} \sum_{s \in \mathcal{S}} \mathbb{E}_{a \sim \pi_h^{t+1}(\cdot|s)} \left[ \frac{\widehat{d}_h^{\mathsf{off}}(s,a)}{\frac{1}{K^{\mathsf{on}}H} + \mathbb{E}_{\pi' \sim \mu}\big[\widehat{d}_h^{\pi'}(s,a)\big]} \right], \tag{20b}$$

where $\eta$ denotes the learning rate to be specified later. We shall discuss how to solve the optimization sub-problem (20b) in Appendix C. The output of this step is a mixture of deterministic policies taking the following form:

$$\pi^{\mathsf{imitate}} = \mathbb{E}_{\pi \sim \mu^{\mathsf{imitate}}}[\pi] \qquad \text{with} \quad \mu^{\mathsf{imitate}} = \frac{1}{T_{\mathsf{max}}} \sum_{t=1}^{T_{\mathsf{max}}} \mu^t. \tag{21}$$

*Exploring the unknown environment.* In addition to mimicking the behavior of the historical dataset, we shall also attempt to explore the environment in a way that complements pre-collected data. Towards this end, it suffices to invoke the reward-agnostic online exploration scheme proposed in Li et al. (2023), whose precise description will be provided in Algorithm 5 in Appendix D.2 to make the paper self-contained. The resulting policy mixture is denoted by

$$\pi^{\mathsf{explore}} = \mathbb{E}_{\pi \sim \mu^{\mathsf{explore}}}[\pi], \tag{22}$$

with $\mu^{\mathsf{explore}} \in \Delta(\Pi)$ representing the associated weight vector.

With the above two exploration policies (21) and (22) in place, we execute the MDP to obtain sample trajectories as follows:

1) Execute the MDP $K_{\mathsf{imitate}}^{\mathsf{on}}$ times using policy $\pi^{\mathsf{imitate}}$ to obtain a dataset containing $K_{\mathsf{imitate}}^{\mathsf{on}} = K^{\mathsf{on}}/3$ independent sample trajectories, denoted by $\mathcal{D}_{\mathsf{imitate}}^{\mathsf{on}}$;

2) Execute the MDP $K_{\mathsf{explore}}^{\mathsf{on}}$ times using policy $\pi^{\mathsf{explore}}$ to obtain a dataset containing $K_{\mathsf{explore}}^{\mathsf{on}} = K^{\mathsf{on}}/3$ independent sample trajectories, denoted by $\mathcal{D}_{\mathsf{explore}}^{\mathsf{on}}$.

**Stage 3: policy learning via offline RL.** Once the above online exploration process is completed, we are positioned to compute a near-optimal policy on the basis of the data in hand. More precisely,

- Let us look at the following dataset

$$\mathcal{D} = \mathcal{D}^{\mathsf{off},2} \cup \mathcal{D}_{\mathsf{imitate}}^{\mathsf{on}} \cup \mathcal{D}_{\mathsf{explore}}^{\mathsf{on}}. \tag{23}$$

In light of the complicated statistical dependency between $\mathcal{D}^{\mathsf{off},1}$ and $\mathcal{D}_{\mathsf{imitate}}^{\mathsf{on}} \cup \mathcal{D}_{\mathsf{explore}}^{\mathsf{on}}$, we only include the second half $\mathcal{D}^{\mathsf{off},2}$ of the offline dataset $\mathcal{D}^{\mathsf{off}}$, so as to exploit the fact that $\mathcal{D}^{\mathsf{off},2}$ is statistically independent from $\mathcal{D}_{\mathsf{imitate}}^{\mathsf{on}} \cup \mathcal{D}_{\mathsf{explore}}^{\mathsf{on}}$.

- We invoke the pessimistic model-based offline RL algorithm proposed in Li et al. (2022) to compute the final policy estimate $\widehat{\pi}$; see Algorithm 6 in Appendix D.3 for more details.

# 4 Main results

As it turns out, Algorithm 1 is capable of achieving provable sample efficiency, as demonstrated in the following theorem. Here and below, we recall that $K = K^{\mathsf{off}} + K^{\mathsf{on}}$.

**Theorem 1.** *Consider $\delta \in (0,1)$ and $\varepsilon \in (0, H]$. Choose the algorithmic parameters such that*

$$\eta = \sqrt{\frac{\log A}{2T_{\mathsf{max}}(K^{\mathsf{on}}H)^2}} \qquad and \qquad T_{\mathsf{max}} \geq 2(K^{\mathsf{on}}H)^2 \log A.$$

*Suppose that*

$$K^{\mathsf{on}} + K^{\mathsf{off}} \geq c_1 \frac{H^3 S C^\star(\sigma)}{\varepsilon^2} \log^2 \frac{K}{\delta} \tag{24a}$$

$$K^{\mathsf{on}} \geq c_1 \frac{H^3 S A \min\{H\sigma, 1\}}{\varepsilon^2} \log \frac{K}{\delta} \tag{24b}$$

*for some large enough constant $c_1 > 0$. Then with probability at least $1 - \delta$, the policy $\widehat{\pi}$ returned by Algorithm 1 satisfies*

$$V_1^\star(\rho) - V^{\widehat{\pi}}(\rho) \leq \varepsilon,$$

*provided that $K^{\mathsf{on}}$ and $K^{\mathsf{off}}$ both exceed some polynomial $\mathsf{poly}(H, S, A, C^\star(\sigma), \log \frac{K}{\delta})$ (independent of $\varepsilon$).*

The proof is deferred to Appendix E. In a nutshell, Theorem 1 uncovers that our algorithm yields $\varepsilon$-accuracy as long as

$$K^{\mathsf{on}} + K^{\mathsf{off}} \gtrsim \frac{H^3 S C^\star(\sigma)}{\varepsilon^2} \log^2 \frac{K}{\delta}, \tag{25a}$$

$$K^{\mathsf{on}} \gtrsim \frac{H^3 S A \min\{H\sigma, 1\}}{\varepsilon^2} \log \frac{K}{\delta}, \tag{25b}$$

ignoring lower-order terms. Several implications of this result are as follows.

**Sample complexity benefits compared with pure online or pure offline RL.** To make apparent its advantage compared with both pure offline and online RL, we make comparisons with several most relevant works. Discussions of other related works are deferred to Appendix A.

*Sample complexity with balanced online and offline data.* For the ease of presentation, let us look at a simple case where $K^{\mathsf{off}} = K^{\mathsf{on}} = K/2$. The the sample complexity bound (25) in this case simplifies to

$$\widetilde{O}\left(\min_{\sigma \in [0,1]} \left\{\frac{H^3 S A \min\{H\sigma, 1\}}{\varepsilon^2} + \frac{H^3 S C^\star(\sigma)}{\varepsilon^2}\right\}\right) =: \widetilde{O}\left(\min_{\sigma \in [0,1]} f_{\mathsf{mixed}}(\sigma)\right). \tag{26}$$

*Comparisons with pure online RL.* We now look at pure online RL, corresponding to the case where $K = K^{\mathsf{on}}$ (so that all sample episodes are collected via online exploration). In this case, the minimax-optimal sample complexity for computing an $\varepsilon$-optimal policy is known to be (Azar et al., 2017; Li et al., 2023)

$$\widetilde{O}\left(\frac{H^3 S A}{\varepsilon^2}\right) = \widetilde{O}\left(f_{\mathsf{mixed}}(1)\right) \tag{27}$$

assuming that $\varepsilon$ is sufficiently small, which is clearly worse than (26). For instance, if there exists some very small $\sigma \ll 1/H$ obeying $C^\star(\sigma) \lesssim 1$, then the ratio of (26) to (27) is at most

$$H\sigma + 1/A \ll 1, \tag{28}$$

thus resulting in substantial sample size savings.

*Comparisons with pure offline RL.* In contrast, in the pure offline case where $K = K^{\mathsf{off}}$, the minimax sample complexity is known to be (Li et al., 2022)

$$\widetilde{O}\left(\frac{H^3 S C^\star(0)}{\varepsilon^2}\right) = \widetilde{O}\left(f_{\mathsf{mixed}}(0)\right) \tag{29}$$

for any target accuracy level $\varepsilon$, which is apparently larger than (26) in general. In particular, recognizing that $C^\star(0) = \infty$ in the presence of incomplete coverage of the state-action space reachable by $\pi^\star$, we might harvest enormous sample size benefits (by exploiting the ability of online RL to visit the previously uncovered state-action-step tuples).

**Comparison with Wagenmaker and Pacchiano (2022).** It is worth noting that Wagenmaker and Pacchiano (2022) also considered policy fine-tuning and proposed a method called FTPedel to tackle linear MDPs. The results therein, however, were mainly instance-dependent, thus making it difficult to compare in general. That being said, we would like to clarify two points:

- Wagenmaker and Pacchiano (2022) imposed all-policy concentrability assumptions, requiring the combined dataset (i.e., the offline and online data altogether) to cover certain feature vectors for all linear softmax policies (see Wagenmaker and Pacchiano (2022, Definition 4.1)). In contrast, our results only assume single-policy (partial) concentrability, which is much weaker than the all-policy counterpart.

- When specializing Wagenmaker and Pacchiano (2022, Corollary 1) to the tabular cases, the sample complexity therein becomes $\widetilde{O}(H^7 S^2 A^2 / \varepsilon^2)$, which is much larger than our result.

**Miscellaneous properties of the proposed algorithm.** In addition to the sample complexity advantages, the proposed hybrid RL enjoys several attributes that could be practically appealing.

*Adaptivity to unknown optimal $\sigma$.* While we have introduced the parameter $\sigma$ to capture incomplete coverage, our algorithm does not rely on any knowledge of $\sigma$. Take the balanced case described around (26) for instance: our algorithm automatically identifies the optimal $\sigma$ that minimizes the function $f_{\mathsf{mixed}}(\sigma)$ over all $\sigma \in [0, 1]$. In other words, Algorithm 1 is able to automatically identify the optimal trade-offs between distribution mismatch and inadequate coverage.

*Reward-agnostic data collection.* It is noteworthy that the online exploration procedure employed in Algorithm 1 does not require any prior information about the reward function. In other words, it is mainly designed to improve coverage of the state-action space, a property independent from the reward function. In truth, the reward function is only queried at the last step to output the learned policy. This enables us to perform hybrid RL in a reward-agnostic manner, which is particularly intriguing in practice, as there is no shortage of scenarios where the reward functions might be engineered subsequently to meet different objectives.

*Strengthening behavior cloning.* Another notable feature is that our algorithm does not rely on prior knowledge about the policies generating the offline dataset $\mathcal{D}^{\mathsf{off}}$; in fact, it is capable of finding a mixed exploration policy $\pi^{\mathsf{imitate}}$ that inherits the advantages of the unknown behavior policy $\pi^{\mathsf{off}}$. This could be of particular interest for behavior cloning, where the offline dataset $\mathcal{D}^{\mathsf{off}}$ is generated by an expert policy, with $C^\star = C^\star(0) \approx 1$, i.e. the expert policy covers the optimal one. In this situation, the supplement of online data collection improves behavior cloning by lowering the statistical error from $\sqrt{\frac{H^3 S C^\star}{K_{\mathsf{off}}}}$ to $\sqrt{\frac{H^3 S C^\star}{K_{\mathsf{off}} + K_{\mathsf{on}}}}$, together with an executable learned policy $\pi^{\mathsf{imitate}}$.

*Computational complexity.* We now take a moment to discuss the computational cost of the proposed algorithm. In Stage 1, we need to first estimate the transition matrices $\{P_h\}$, which can be accomplished with runtime $O(K^{\mathsf{on}})$. In the ensuing stages, we call Algorithm 3 to estimate $\widehat{d^\pi}$ for each $\pi$ we encounter. When computing $\pi^{\mathsf{imitate}}$, we need to calculate $\widehat{d^\pi}$ for $T_1 = T_{\mathsf{max}} T_2$ times, where $T_2$ denotes the number of iterations for calculating $\mu^{t+1}$ in Eq. (20b); in comparison, the computational cost of estimating $d^\pi$ to yield $\pi^{\mathsf{explore}}$ in Eq. (22) is much smaller. For each $\widehat{d^\pi}$, it needs $O(HS^2A)$ computation. With a slight modification on the target Eq. (19) as follows

$$
\mu^{\mathsf{imitate}} \approx
$$
$$
\arg\min_{\mu \in \Delta(\Pi)} \sum_{h=1}^{H} \sum_{s \in \mathcal{S}} \max_{a \in \mathcal{A}} \frac{\widehat{d}_h^{\mathsf{off}}(s,a)}{\frac{1}{KH} + O\left(\frac{1}{SH}\right)\widehat{d}_h^{\mathsf{off}}(s,a) + \underset{\pi' \sim \mu^{\mathsf{explore}}}{\mathbb{E}}\left[\widehat{d}_h^{\pi'}(s,a)\right] + \underset{\pi' \sim \mu}{\mathbb{E}}\left[\widehat{d}_h^{\pi'}(s,a)\right]},
$$
(30)

we can find a good enough $\pi^{\mathsf{imitate}}$ with $T_{\mathsf{max}} = \widetilde{O}(H^2 S^2)$ for $\eta \asymp \frac{1}{H^2 S^2}$ and $O(H^4 S^4 A^2)$ Frank-Wolfe updates for $\alpha \asymp \frac{1}{H^3 S^3 A^2}$. These taken collectively lead to the following computational complexity for each stage: $O(K^{\mathsf{on}} + H^7 S^8 A^3 + K^{\mathsf{off}} H)$ for Stage 1, $O(H^7 S^7 A^3)$ for Stage 2, and $O(KH)$ for Stage 3.

# 5  Discussion

We have studied the policy fine-tuning problem of practical interest, where one is allowed to exploit pre-collected historical data to facilitate and improve online RL. We have proposed a three-stage algorithm tailored to the tabular setting, which attains provable sample size savings compared with both pure online RL and pure offline RL algorithms. Our algorithm design has leveraged key insights from recent advances in both model-based offline RL and reward-agnostic online RL.

While the proposed algorithm achieves provable sample efficiency, this cannot be guaranteed unless the sample size already surpasses a fairly large threshold (in other words, the algorithm imposes a high burn-in cost). It would be of great interest to see whether one can achieve sample optimality for the entire $\varepsilon$-range. Another issue arises from the computation side: even though the proposed algorithm can be implemented in polynomial time, the computational complexity of the Frank-Wolfe-type subroutine might already be too expensive for solving problems with enormous dimensionality. Can we hope to further accelerate it to make it practically more appealing? Finally, it might also be interesting to study sample-efficient hybrid RL in the presence of low-complexity function approximation, in the hope of further reducing sample complexity.

## Acknowledgements

Y. Chen is supported in part by the Alfred P. Sloan Research Fellowship, the Google Research Scholar Award, the AFOSR grant FA9550-22-1-0198, the ONR grant N00014-22-1-2354, and the NSF grants CCF-2221009 and CCF-1907661. Y. Chi are supported in part by the grants ONR N00014-19-1-2404, NSF CCF-2106778, DMS-2134080 and CNS-2148212. J. D. Lee acknowledges support of the ARO under MURI Award W911NF-11-1-0304, the Sloan Research Fellowship, the NSF grants CCF 2002272, IIS 2107304 and CIF 2212262, the ONR Young Investigator Award, and the NSF CAREER Award 2144994.

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

# A  Other related works

In this section, we briefly discuss a small set of additional prior works related to the current paper.

**(Reward-aware) online RL.**  In online RL, an agent seeks to find a near-optimal policy by sequentially and adaptively interacting with the unknown environment, without having access to any additional offline dataset. The extensive studies of online RL gravitate around how to optimally trade off exploration against exploitation, for which the principle of optimism in the face of uncertainty plays a crucial role (Auer and Ortner, 2006; Jaksch et al., 2010; Azar et al., 2017; Dann et al., 2017; Jin et al., 2018; Bai et al., 2019; Dong et al., 2019; Zhang et al., 2020b; Ménard et al., 2021b; Li et al., 2021b; Zhang et al., 2023). Information-theoretic regret lower bounds have been established by Domingues et al. (2021); Jin et al. (2018), which are shown to be achievable (up to log factor) by the model-based approach for arbitrary sample sizes (Zhang et al., 2023). A further strand of works extended these studies to the case with function approximation, including both linear function approximation (Jin et al., 2020b; Zanette et al., 2020; Zhou et al., 2021a; Li et al., 2021a) and other more general families of function approximation (Du et al., 2021; Jin et al., 2021a; Foster et al., 2021).

**Offline RL.**  In contrast to online RL, offline RL assumes access to a pre-collected offline dataset and precludes active interactions with the environment. Given the absence of further data collection, the sample complexity of pure offline RL depends heavily upon the quality of the offline dataset at hand, which has often been characterized via some sorts of concentrability coefficients in prior works (Rashidinejad et al., 2021; Zhan et al., 2022). Earlier works (Munos and Szepesvári, 2008; Chen and Jiang, 2019) typically operated under the assumption of all-policy concentrability — namely, the assumption that the dataset covers the visited state-action pairs of all possible policies — thus imposing a stringent requirement for the offline dataset to be highly explorative. To circumvent this stringent assumption, Liu et al. (2020); Kumar et al. (2020); Jin et al. (2021b); Rashidinejad et al. (2021); Uehara and Sun (2021); Li et al. (2022); Yin et al. (2021); Shi et al. (2022); Yan et al. (2023) incorporated the pessimism principle amid uncertainty into the algorithm designs and, as a result, required only single-policy concentrability (so that the dataset only needs to cover the part of the state-action space reachable by the optimal policy). With regards to the basic tabular case, Li et al. (2022) proved that the pessimistic model-based offline algorithm is capable of achieving minimax-optimal sample complexity for the full $\varepsilon$-range, accommodating both the episodic finite-horizon case and the discounted infinite-horizon analog. Moving beyond single-agent tabular settings, a recent line of works investigated offline RL in the presence of general function approximation (Jin et al., 2020c; Xie et al., 2021a; Zhan et al., 2022), environment shift (Zhou et al., 2021b; Shi and Chi, 2022), and in the context of zero-sum Markov games (Cui and Du, 2022; Yan et al., 2022).

**Hybrid RL.**  While there were a number of empirical works (Rajeswaran et al., 2017; Vecerik et al., 2017; Kalashnikov et al., 2018; Hester et al., 2018; Nair et al., 2018, 2020) suggesting the perfromance gain of combining online RL with offline datasets (compared to pure online or offline learning), rigorous theoretical evidence remained highly limited. Ross and Bagnell (2012); Xie et al. (2021b) attempted to develop theoretical understanding by looking at one special hybrid scenario, where the agent can perform either of the following in each episode: (i) collecting a new online episode; and (ii) executing a prescribed and fixed reference policy to generate a sample episode. In this setting, Xie et al. (2021b) showed that in the minimax sense, combining online learning with samples generated by such a reference policy is not advantageous in comparison with pure online or offline RL. Note that our results do not contradict with the lower bound in Xie et al. (2021b), given that we exploit "partial" single-policy concentrability that implies additional structure except for the worst case. Akin to the current paper, Song et al. (2022); Wagenmaker and Pacchiano (2022) studied online RL with additional access to an offline dataset. Nevertheless, Song et al. (2022) mainly focused on the issue of computational efficiency, and the algorithm proposed therein does not come with improved sample complexity. In contrast, Wagenmaker and Pacchiano (2022) focused attention on statistical efficiency, although the sample complexity derived therein is highly suboptimal when specialized to the tabular setting.

**Reward-free and task-agnostic exploration.**  Reward-free and task-agnostic exploration, which refer to the scenario where the agent first collects online sample trajectories without guidance of any information about the reward function(s), has garnered much recent attention (Brafman and

Tennenholtz, 2002; Jin et al., 2020a; Zhang et al., 2020a, 2021b; Huang et al., 2022). Focusing on the tabular case, the earlier work Jin et al. (2020a) put forward a reward-free exploration scheme that achieves minimax optimality in terms of the dependency on $S$, $A$ and $1/\varepsilon$, with the horizon dependency further improved by subsequent works (Kaufmann et al., 2021; Ménard et al., 2021a; Li et al., 2023). In particular, the exploration scheme proposed in Li et al. (2023) was shown to achieve minimax-optimal sample complexity when there exist a polynomial number of pre-determined but unseen reward functions of interest, which inspires the algorithm design of the present paper. More-over, reward-free RL has been extended to account for function approximation, including both linear (Wang et al., 2020; Agarwal et al., 2020; Qiao and Wang, 2022; Zhang et al., 2021a; Wagenmaker et al., 2022) and nonlinear function classes (Chen et al., 2022a).

# B Pseudocode of Algorithm 1

In this section, we provide the whole procedure for the proposed hybrid RL algorithm, with several subroutines deferred to Appendix C and Appendix D.

---

**Algorithm 1:** The proposed hybrid RL algorithm.

---

1 **Input:** offline dataset $\mathcal{D}^{\mathsf{off}}$ (containing $K^{\mathsf{off}}$ trajectories), parameters $N$, $K^{\mathsf{on}}$, $T_{\mathsf{max}}$, learning rate $\eta$.

2 **Initialize:** $\pi_h^1(a\,|\,s) = 1/A$ for any $(s,a,h)$; $K = K^{\mathsf{off}} + K^{\mathsf{on}}$; split $\mathcal{D}^{\mathsf{off}}$ into two halves $\mathcal{D}^{\mathsf{off},1}$ and $\mathcal{D}^{\mathsf{off},2}$.

/* Estimation of occupancy distributions for any policy $\pi$. */

3 Call Algorithm 3, which allows one to specify $\widehat{d}_h^\pi(s,a)$ for any deterministic policy $\pi$ and any $(s,a,h)$.

/* Estimation of occupancy distributions of the historical data. */

4 Use the dataset $\mathcal{D}^{\mathsf{off},1}$ to compute

$$
\widehat{d}_h^{\mathsf{off}}(s,a) = \frac{2N_h^{\mathsf{off}}(s,a)}{K^{\mathsf{off}}} \mathbb{1}\left( \frac{N_h^{\mathsf{off}}(s,a)}{K^{\mathsf{off}}} \geq c_{\mathsf{off}}\left\{ \frac{\log\frac{HSA}{\delta}}{K^{\mathsf{off}}} + \frac{H^4 S^4 A^4 \log\frac{HSA}{\delta}}{N} + \frac{SA}{K^{\mathsf{on}}} \right\} \right)
$$

for any $(s,a,h)$, where $N_h^{\mathsf{off}}(s,a) = \sum_{k=1}^{K^{\mathsf{off}}/2} \mathbb{1}(s_h^k = s, a_h^k = a)$ and $c_{\mathsf{off}} > 0$ is some absolute constant.

/* Compute a general sample-efficient online exploration scheme. */

5 Call Algorithm 5 with estimators $\widehat{d}^\pi$ to compute policy $\pi^{\mathsf{explore}}$ and the associated weight $\mu^{\mathsf{explore}}$.

/* Compute an online exploration scheme tailored to the offline dataset. */

6 **for** $t = 1, \cdots, T_{\mathsf{max}}$ **do**

7     Compute $\mu^t$ using Algorithm 2.

8     Update $\pi_h^{t+1}(a\,|\,s)$ for all $(s,a,h) \in \mathcal{S} \times \mathcal{A} \times [H]$ such that:

$$
\pi_h^{t+1}(a\,|\,s) = \frac{\exp\left( \eta \sum_{k=1}^t \frac{\widehat{d}_h^{\mathsf{off}}(s,a)}{\frac{1}{K^{\mathsf{on}}H} + \mathbb{E}_{\pi' \sim \mu^k}\left[ \widehat{d}_h^{\pi'}(s,a) \right]} \right)}{\sum_{a' \in \mathcal{A}} \exp\left( \eta \sum_{k=1}^t \frac{\widehat{d}_h^{\mathsf{off}}(s,a')}{\frac{1}{K^{\mathsf{on}}H} + \mathbb{E}_{\pi' \sim \mu^k}\left[ \widehat{d}_h^{\pi'}(s,a') \right]} \right)},
$$

9 Set $\mu^{\mathsf{imitate}} = \frac{1}{T_{\mathsf{max}}} \sum_{t=1}^{T_{\mathsf{max}}} \mu^t$ and $\pi^{\mathsf{imitate}} = \mathbb{E}_{\pi \sim \mu^{\mathsf{imitate}}}[\pi]$.

/* Sampling using the above two exploration policies. */

10 Collect $K_{\mathsf{imitate}}^{\mathsf{on}}$ (resp. $K_{\mathsf{explore}}^{\mathsf{on}}$) sample trajectories using $\pi^{\mathsf{imitate}}$ (resp. $\pi^{\mathsf{explore}}$) to form a dataset $\mathcal{D}_{\mathsf{imitate}}^{\mathsf{on}}$ (resp. $\mathcal{D}_{\mathsf{explore}}^{\mathsf{on}}$).

/* Run the model-based offline RL algorithm. */

11 Apply Algorithm 6 to the dataset $\mathcal{D} = \mathcal{D}^{\mathsf{off},2} \cup \mathcal{D}_{\mathsf{imitate}}^{\mathsf{on}} \cup \mathcal{D}_{\mathsf{explore}}^{\mathsf{on}}$ to compute a policy $\widehat{\pi}$.

12 **Output:** policy $\widehat{\pi}$.

---

## C Subroutine for solving the subproblem (20b)

While (20b) is a convex optimization subproblem, it involves optimization over a parameter space with exponentially large dimensions. In order to solve it in a computationally feasible manner, we propose a tailored subroutine based on the Frank-Wolfe algorithm (Bubeck, 2015).

Before proceeding, recall that when specifying $\widehat{d}^\pi$ in Algorithm 3, we draw $N$ independent trajectories $\{s_1^{n,h}, a_1^{n,h}, \ldots, s_{h+1}^{n,h}\}_{1 \leq n \leq N}$ and compute an empirical estimate $\widehat{P}_h$ of the probability transition kernel at step $h$ such that

$$\widehat{P}_h(s' \mid s, a) = \frac{\mathbb{1}(N_h(s,a) > \xi)}{\max\{N_h(s,a), 1\}} \sum_{n=1}^{N} \mathbb{1}(s_h^{n,h} = s, a_h^{n,h} = a, s_{h+1}^{n,h} = s'), \qquad \forall(s,a,s') \in \mathcal{S} \times \mathcal{A} \times \mathcal{S}, \tag{31}$$

where $N_h(s,a) = \sum_{n=1}^{N} \mathbb{1}\{s_h^{n,h} = s, a_h^{n,h} = a\}$.

**The proposed Frank-Wolfe-type algorithm.** With this set of notation in place and with an initial guess taken to be the indicator function $\mu^{(1)} = \mathbb{1}_{\pi_{\mathsf{init}}}$ for an arbitrary policy $\pi_{\mathsf{init}} \in \Pi$, the $k$-th iteration of our iterative procedure for solving (20b) can be described as follows.

- *Computing a search direction.* The search direction of the Frank-Wolfe algorithm is typically taken to be a feasible direction that maximizes its correlation with the gradient of the objective function (Bubeck, 2015). When specialized to the current sub-problem (20b), the search direction can be taken to be the Dirac measure $\delta_{\pi^{(k)}}$, where

$$\pi^{(k)} = \arg\max_{\pi \in \Pi} f\left(\pi, \mu^{(k)}\right) := \sum_{h=1}^{H} \sum_{s \in \mathcal{S}} \mathbb{E}_{a \sim \pi_h^{t+1}(\cdot|s)} \left[ \frac{\widehat{d}_h^\pi(s,a) \widehat{d}_h^{\mathsf{off}}(s,a)}{\left(\frac{1}{K^{\mathsf{on}}H} + \mathbb{E}_{\pi' \sim \mu^{(k)}}\left[\widehat{d}_h^{\pi'}(s,a)\right]\right)^2} \right]. \tag{32}$$

  As it turns out, this optimization problem (32) can be efficiently solved by applying dynamic programming (Bertsekas, 2017) to an augmented MDP $\mathcal{M}^{\mathsf{off}}$ constructed as follows.

  - Introduce an augmented finite-horizon MDP $\mathcal{M}^{\mathsf{off}} = (\mathcal{S} \cup \{s_{\mathsf{aug}}\}, \mathcal{A}, H, \widehat{P}^{\mathsf{aug}}, r^{\mathsf{off}})$, where $s_{\mathsf{aug}}$ is an augmented state. We choose the reward function to be

$$r_h^{\mathsf{off}}(s,a) = \begin{cases} \frac{\pi_h^{t+1}(a \mid s) \widehat{d}_h^{\mathsf{off}}(s,a)}{\left(\frac{1}{K^{\mathsf{on}}H} + \mathbb{E}_{\pi' \sim \mu^{(k)}}\left[\widehat{d}_h^{\pi'}(s,a)\right]\right)^2}, & \text{if } (s,a,h) \in \mathcal{S} \times \mathcal{A} \times [H], \\ 0, & \text{if } (s,a,h) \in \{s_{\mathsf{aug}}\} \times \mathcal{A} \times [H], \end{cases} \tag{33}$$

  and the probability transition kernel as

$$\widehat{P}_h^{\mathsf{aug}}(s' \mid s, a) = \begin{cases} \widehat{P}_h(s' \mid s, a), & \text{if } s' \in \mathcal{S} \\ 1 - \sum_{s' \in \mathcal{S}} \widehat{P}_h(s' \mid s, a), & \text{if } s' = s_{\mathsf{aug}} \end{cases} \quad \text{for all } (s,a,h) \in \mathcal{S} \times \mathcal{A} \times [H], \tag{34a}$$

$$\widehat{P}_h^{\mathsf{aug}}(s' \mid s_{\mathsf{aug}}, a) = \mathbb{1}(s' = s_{\mathsf{aug}}) \qquad \text{for all } (a,h) \in \mathcal{A} \times [H]. \tag{34b}$$

- *Frank-Wolfe updates.* We then update the iterate $\mu^{(k+1)}$ as a convex combination of the current iterate and the direction found in the previous step:

$$\mu^{(k+1)} = (1 - \alpha)\mu^{(k)} + \alpha \mathbb{1}_{\pi^{(k)}}, \tag{35}$$

  where the stepsize is chosen to be

$$\alpha = \frac{S}{(K^{\mathsf{on}}H)^3}. \tag{36}$$

**Stopping rule.** It is also necessary to specify the stopping rule of the above iterative procedure. Throughout this paper, the above subroutine will terminate as long as

$$\sum_{h=1}^{H} \sum_{s \in \mathcal{S}} \mathbb{E}_{a \sim \pi_h^{t+1}(\cdot \mid s)} \left[ \frac{\widehat{d}_h^{\mathsf{off}}(s, a)}{\frac{1}{K^{\mathsf{on}} H} + \mathbb{E}_{\pi' \sim \mu^{(k)}} \left[ \widehat{d}_h^{\pi'}(s, a) \right]} \right] \leq 108 SH, \tag{37}$$

with the final output taken to be $\mu^{t+1} = \mu^{(k)}$. We shall justify the feasibility of this stopping rule (namely, the fact that this stopping criterion can be met by some mixed policy) in Appendix F.

**Iteration complexity.** Encouragingly, the above subroutine in conjunction with the stopping rule (37) leads to an iteration complexity no larger than

$$\text{(iteration complexity)} \qquad O\left( \frac{(K^{\mathsf{on}} H)^4}{S^2} \right) \tag{38}$$

The proof of this claim is postponed to Section F.

---

**Algorithm 2:** Subroutine for solving the sub-problem (20b).

---

1 **Initialize:** $\mu^{(1)} = \mathbb{1}_{\pi_{\mathsf{init}}}$ for an arbitrary policy $\pi_{\mathsf{init}} \in \Pi$.
2 **for** $k = 1, 2, \cdots$ **do**
3     Exit for-loop if the following condition is met: // `stopping criterion`
4

$$\sum_{h=1}^{H} \sum_{s \in \mathcal{S}} \mathbb{E}_{a \sim \pi_h^{t+1}(\cdot \mid s)} \left[ \frac{\widehat{d}_h^{\mathsf{off}}(s, a)}{\frac{1}{K^{\mathsf{on}} H} + \mathbb{E}_{\pi' \sim \mu^{(k)}} \left[ \widehat{d}_h^{\pi'}(s, a) \right]} \right] \leq 108 SH. \tag{39}$$

    /* `Find the search direction` */
5     Compute the optimal deterministic policy $\pi^{(k),\mathsf{aug}}$ of the MDP
    $\mathcal{M}_{\mathsf{off}} = (\mathcal{S} \cup \{s_{\mathsf{aug}}\}, \mathcal{A}, H, \widehat{P}^{\mathsf{aug}}, r_{\mathsf{off}})$, where $s_{\mathsf{aug}}$ is an augmented state,

$$r_h^{\mathsf{off}}(s, a) = \begin{cases} \frac{\pi_h^{t+1}(a \mid s) \widehat{d}_h^{\mathsf{off}}(s, a)}{\left( \frac{1}{K^{\mathsf{on}} H} + \mathbb{E}_{\pi' \sim \mu^{(k)}} \left[ \widehat{d}_h^{\pi'}(s, a) \right] \right)^2}, & \text{if } (s, a, h) \in \mathcal{S} \times \mathcal{A} \times [H], \\ 0, & \text{if } (s, a, h) \in \{s_{\mathsf{aug}}\} \times \mathcal{A} \times [H], \end{cases} \tag{40}$$

    and the augmented probability transition kernel is given by

$$\widehat{P}_h^{\mathsf{aug}}(s' \mid s, a) = \begin{cases} \widehat{P}_h(s' \mid s, a), & \text{if } s' \in \mathcal{S} \\ 1 - \sum_{s' \in \mathcal{S}} \widehat{P}_h(s' \mid s, a), & \text{if } s' = s_{\mathsf{aug}} \end{cases} \quad \text{for all } (s, a, h) \in \mathcal{S} \times \mathcal{A} \times [H];$$

(41a)

$$\widehat{P}_h^{\mathsf{aug}}(s' \mid s_{\mathsf{aug}}, a) = \mathbb{1}(s' = s_{\mathsf{aug}}) \qquad\qquad \text{for all } (a, h) \in \mathcal{A} \times [H].$$

(41b)

6     Let $\pi^{(k)}$ be the corresponding optimal deterministic policy of $\pi^{(k),\mathsf{aug}}$ in the original state space.
7     Update // `Frank-Wolfe update`
8

$$\mu^{(k+1)} = (1 - \alpha) \mu^{(k)} + \alpha \mathbb{1}_{\pi^{(k)}}, \quad \text{where} \quad \alpha = \frac{S}{(K^{\mathsf{on}} H)^3}.$$

9 **Output:** the policy mixture $\mu^{t+1} = \mu^{(k)}$.

---

## D Useful algorithmic subroutines from prior works

In this section, we provide precise descriptions of several useful algorithmic subroutines that have been developed in recent works. The algorithm procedures are directly quoted from these prior works, with slight modification.

### D.1 Subroutine: occupancy estimation for any policy $\pi$

The first subroutine we'd like to describe is concerned with estimating the occupancy distribution $d^\pi$ induced by any policy $\pi$, based on a carefully designed exploration strategy. This algorithm, proposed by Li et al. (2023), seeks to estimate $\{d_h^\pi\}$ step by step (i.e., from $h = 1, \dots, H$). For each $h$, it computes an appropriate exploration policy $\pi^{\mathsf{explore},h}$ to adequately explore what happens between step $h$ and step $h + 1$, and then collect $N$ sample trajectories using $\pi^{\mathsf{explore},h}$. These turns allow us to estimate the occupancy distribution $d_{h+1}^\pi$ for step $h + 1$. See Algorithm 3 for a precise description.

---

**Algorithm 3:** Subroutine for estimating occupancy distributions for any policy $\pi$ (Li et al., 2023).

---

1 **Input:** target success probability $1 - \delta$, threshold $\xi = c_\xi H^3 S^3 A^3 \log(HSA/\delta)$.
   /* Estimate occupancy distributions for step 1. */
2 Draw $N$ independent episodes (using arbitrary policies), whose initial states are i.i.d. drawn
   from $s_1^{n,0} \overset{\text{i.i.d.}}{\sim} \rho$ $(1 \le n \le N)$. Define the following functions

$$\widehat{d}_1^\pi(s) = \frac{1}{N} \sum_{n=1}^N \mathbb{1}\{s_1^{n,0} = s\}, \qquad \widehat{d}_1^\pi(s,a) = \widehat{d}_1^\pi(s)\pi_1(a \,|\, s) \tag{42}$$

   for any deterministic policy $\pi : \mathcal{S} \times [H] \to \Delta(\mathcal{A})$ and any $(s,a) \in \mathcal{S} \times \mathcal{A}$. (Note that these functions are defined for future use and not computed for the moment, as we have not specified policy $\pi$.)
   /* Estimate occupancy distributions for steps $2, \dots, H$. */
3 **for** $h = 1$ **to** $H - 1$ **do**
     /* Collect $N$ sample trajectories using a suitable exploration policy. */
4     Call Algorithm 4 to compute an exploration policy $\pi^{\mathsf{explore},h}$ and compute an estimate $\widehat{P}_h$ of the true transition kernel $P_h$.
     /* Specify how to compute $\widehat{d}_{h+1}^\pi$ for any policy $\pi$. */
5     For any deterministic policy $\pi : \mathcal{S} \times [H] \to \Delta(\mathcal{A})$ and any $(s,a) \in \mathcal{S} \times \mathcal{A}$, define
$$\widehat{d}_{h+1}^\pi(s) = \langle \widehat{P}_h(s \,|\, \cdot, \cdot), \, \widehat{d}_h^\pi(\cdot, \cdot) \rangle, \qquad \widehat{d}_{h+1}^\pi(s,a) = \widehat{d}_{h+1}^\pi(s)\pi_{h+1}(a \,|\, s). \tag{43}$$

---

We note, however, that Algorithm 3 requires another subroutine to compute a suitable exploration policy $\pi^{\mathsf{explore},h}$. As it turns out, this can be accomplished by approximately solving the following problem

$$\widehat{\mu}^h \approx \arg \max_{\mu \in \Delta(\Pi)} \sum_{(s,a) \in \mathcal{S} \times \mathcal{A}} \log \left[ \frac{1}{KH} + \underset{\pi \sim \mu}{\mathbb{E}} \left[ \widehat{d}_h^\pi(s,a) \right] \right] \tag{44}$$

via the Frank-Wolfe algorithm and returning $\pi^{\mathsf{explore},h} = \mathbb{E}_{\pi \sim \widehat{\mu}^h}[\pi]$. See Algorithm 4 for details.

### D.2 Subroutine: reward-agnostic online exploration

Based on the estimated occupancy distributions specified in Algorithm 3, Li et al. (2023) proposed a reward-independent online exploration scheme that proves useful in exploring an unknown environment. In a nutshell, this scheme computes a desired exploration policy by approximately solving the following optimization sub-problem:

$$\mu^{\mathsf{explore}} \approx \arg \max_{\mu \in \Delta(\Pi)} \left\{ \sum_{h=1}^H \sum_{(s,a) \in \mathcal{S} \times \mathcal{A}} \log \left[ \frac{1}{K^{\mathsf{on}}H} + \mathbb{E}_{\pi \sim \mu} \left[ \widehat{d}_h^\pi(s,a) \right] \right] \right\}. \tag{47}$$

again using the Frank-Wolfe algorithm; the resulting policy takes the form of a mixture of deterministic policies, as given by $\pi^{\mathsf{explore}} = \mathbb{E}_{\pi \sim \mu^{\mathsf{explore}}}[\pi]$. This exploration policy is then employed to execute the MDP for a number of times in order to collect enough information about the unknowns. See Algorithm 5 for the whole procedure.

**Algorithm 4:** Subroutine for computing the exploration policy for step $h$ in occupancy estimation (Li et al., 2023).

---

**1 Initialize:** $\mu^{(0)} = \mathbb{1}_{\pi_{\mathsf{init}}}$ for an arbitrary policy $\pi_{\mathsf{init}} \in \Pi$, $T_{\max} = \lfloor 50SA \log(KH) \rfloor$.

**2 for** $t = 0$ to $T_{\max}$ **do**

/* find the optimal policy */

**3** Compute the optimal deterministic policy $\pi^{(t),\mathsf{b}}$ of the augmented MDP $\mathcal{M}_{\mathsf{b}}^h = (\mathcal{S} \cup \{s_{\mathsf{aug}}\}, \mathcal{A}, H, \widehat{P}^{\mathsf{aug},h}, r_{\mathsf{b}}^h)$, where $s_{\mathsf{aug}}$ is an augmented state,

$$r_{\mathsf{b},j}^h(s,a) = \begin{cases} \frac{1}{\frac{1}{K^{\mathsf{on}}H} + \mathbb{E}_{\pi \sim \mu^{(t)}}\left[\widehat{d}_h^\pi(s,a)\right]}, & \text{if } (s,a,j) \in \mathcal{S} \times \mathcal{A} \times \{h\}, \\ 0, & \text{if } s = s_{\mathsf{aug}} \text{ or } j \neq h, \end{cases} \quad (45)$$

and the augmented probability transition kernel is defined as

$$\widehat{P}_j^{\mathsf{aug},h}(s' \mid s, a) = \begin{cases} \widehat{P}_j(s' \mid s, a), & \text{if } s' \in \mathcal{S} \\ 1 - \sum_{s' \in \mathcal{S}} \widehat{P}_j(s' \mid s, a), & \text{if } s' = s_{\mathsf{aug}} \end{cases} \quad \text{for all } (s,a,j) \in \mathcal{S} \times \mathcal{A} \times [h];$$

$$(46a)$$

**4** $\widehat{P}_j^{\mathsf{aug},h}(s' \mid s, a) = \mathbb{1}(s' = s_{\mathsf{aug}}) \qquad\qquad\qquad \text{if } s = s_{\mathsf{aug}} \text{ or } j > h. \quad (46b)$

Let $\pi^{(t)}$ be the corresponding optimal deterministic policy of $\pi^{(t),\mathsf{b}}$ in the original state space.

**5** Compute // choose the stepsize

$$\alpha_t = \frac{\frac{1}{SA}g(\pi^{(t)}, \widehat{d}, \mu^{(t)}) - 1}{g(\pi^{(t)}, \widehat{d}, \mu^{(t)}) - 1}, \quad \text{where} \quad g(\pi, \widehat{d}, \mu) = \sum_{(s,a) \in \mathcal{S} \times \mathcal{A}} \frac{\frac{1}{K^{\mathsf{on}}H} + \widehat{d}_h^\pi(s,a)}{\frac{1}{K^{\mathsf{on}}H} + \mathbb{E}_{\pi \sim \mu}[\widehat{d}_h^\pi(s,a)]}.$$

Here, $\widehat{d}_h^\pi(s,a)$ is computed via (42) for $h = 1$, and (43) for $h \geq 2$.

**7** If $g(\pi^{(t)}, \widehat{d}, \mu^{(t)}) \leq 2SA$ then exit for-loop. // stopping rule

**8** Update // Frank-Wolfe update

**9**
$$\mu^{(t+1)} = (1 - \alpha_t)\,\mu^{(t)} + \alpha_t\,\mathbb{1}_{\pi^{(t)}}.$$

**10** Set $\pi^{\mathsf{explore},h} = \mathbb{E}_{\pi \sim \mu^{(t)}}[\pi]$ with $\widehat{\mu}^h = \mu^{(t)}$. // The final exploration policy for step $h$.

/* Draw samples using $\pi^{\mathsf{explore},h}$ to estimate the transition kernel. */

**11** Draw $N$ independent trajectories $\{s_1^{n,h}, a_1^{n,h}, \ldots, s_{h+1}^{n,h}\}_{1 \leq n \leq N}$ using policy $\pi^{\mathsf{explore},h}$ and compute

$$\widehat{P}_h(s' \mid s, a) = \frac{\mathbb{1}(N_h(s,a) > \xi)}{\max\{N_h(s,a), 1\}} \sum_{n=1}^N \mathbb{1}(s_h^{n,h} = s, a_h^{n,h} = a, s_{h+1}^{n,h} = s'), \qquad \forall (s,a,s') \in \mathcal{S} \times \mathcal{A} \times \mathcal{S},$$

where $N_h(s,a) = \sum_{n=1}^N \mathbb{1}\{s_h^{n,h} = s, a_h^{n,h} = a\}$.

**12 Output:** the exploration policy $\pi^{\mathsf{explore},h}$, the weight $\widehat{\mu}^h$, and the estimated kernel $\widehat{P}_h$.

---

### D.3 Subroutine: pessimistic model-based offline RL

Given a historical dataset containing a collection of statistically independent sample trajectories, Li et al. (2022) came up with a model-based approach that enjoys provable minimax optimality. This approach first employs a two-fold subsampling trick in order to decouple the statistical dependency across different steps of a single trajectory. After this subsampling step, this approach resorts to the principle of pessimism in the face of uncertainty, which employs value iteration but penalizes the updates via proper variance-aware penalization (i.e., Bernstein-style lower confidence bounds). Details can be found in Algorithm 6.

---

**Algorithm 5:** Subroutine for computing the desired online exploration policy (Li et al., 2023).

---

**1 Initialize**: $\mu_{\mathsf{b}}^{(0)} = \delta_{\pi_{\mathsf{init}}}$ for an arbitrary policy $\pi_{\mathsf{init}} \in \Pi$, $T_{\max} = \lfloor 50SAH \log(KH) \rfloor$.

**2 for** $t = 0$ **to** $T_{\max}$ **do**

  `/* find the optimal policy */`

**3**  Compute the optimal deterministic policy $\pi^{(t),\mathsf{b}}$ of the MDP

  $\mathcal{M}_{\mathsf{b}} = (\mathcal{S} \cup \{s_{\mathsf{aug}}\}, \mathcal{A}, H, \widehat{P}^{\mathsf{aug}}, r_{\mathsf{b}})$, where $s_{\mathsf{aug}}$ is an augmented state,

  $$r_{\mathsf{b},h}(s,a) = \begin{cases} \frac{1}{\frac{1}{K^{\mathsf{on}}H} + \mathbb{E}_{\pi \sim \mu_{\mathsf{b}}^{(t)}}\left[\widehat{d}_h^{\pi}(s,a)\right]}, & \text{if } (s,a,h) \in \mathcal{S} \times \mathcal{A} \times [H], \\ 0, & \text{if } (s,a,h) \in \{s_{\mathsf{aug}}\} \times \mathcal{A} \times [H], \end{cases} \tag{48}$$

  and the augmented probability transition kernel is given by

  $$\widehat{P}_h^{\mathsf{aug}}(s' \,|\, s,a) = \begin{cases} \widehat{P}_h(s' \,|\, s,a), & \text{if } s' \in \mathcal{S} \\ 1 - \sum_{s' \in \mathcal{S}} \widehat{P}_h(s' \,|\, s,a), & \text{if } s' = s_{\mathsf{aug}}, \end{cases} \quad \text{for all } (s,a,h) \in \mathcal{S} \times \mathcal{A} \times [H]; \tag{49a}$$

  $$\widehat{P}_h^{\mathsf{aug}}(s' \,|\, s_{\mathsf{aug}},a) = \mathbb{1}(s' = s_{\mathsf{aug}}) \quad \text{for all } (a,h) \in \mathcal{A} \times [H]. \tag{49b}$$

**4**  Let $\pi^{(t)}$ be the corresponding optimal deterministic policy of $\pi^{(t),\mathsf{b}}$ in the original state space.

**5** Compute `// choose the stepsize`
**6**

  $$\alpha_t = \frac{\frac{1}{SAH}g(\pi^{(t)},\widehat{d},\mu_{\mathsf{b}}^{(t)}) - 1}{g(\pi^{(t)},\widehat{d},\mu_{\mathsf{b}}^{(t)}) - 1}, \quad \text{where} \quad g(\pi,\widehat{d},\mu) = \sum_{h=1}^{H} \sum_{(s,a) \in \mathcal{S} \times \mathcal{A}} \frac{\frac{1}{K^{\mathsf{on}}H} + \widehat{d}_h^{\pi}(s,a)}{\frac{1}{K^{\mathsf{on}}H} + \mathbb{E}_{\pi \sim \mu}\left[\widehat{d}_h^{\pi}(s,a)\right]}.$$

  Here, $\widehat{d}_h^{\pi}(s,a)$ is computed via (42) for $h = 1$, and (43) for $h \geq 2$.

**7**  If $g(\pi^{(t)},\widehat{d},\mu_{\mathsf{b}}^{(t)}) \leq 2HSA$ then exit for-loop. `// stopping rule`
**8** Update `// Frank-Wolfe update`
**9**

  $$\mu_{\mathsf{b}}^{(t+1)} = (1 - \alpha_t)\,\mu_{\mathsf{b}}^{(t)} + \alpha_t\,\mathbb{1}_{\pi^{(t)}}\,.$$

**10 Output:** the exploration policy $\pi^{\mathsf{explore}} = \mathbb{E}_{\pi \sim \mu_{\mathsf{b}}^{(t)}}[\pi]$ and the associated weight $\mu^{\mathsf{explore}} = \mu_{\mathsf{b}}^{(t)}$.

---

---

**Algorithm 6:** A pessimistic model-based offline RL algorithm (Li et al., 2022).

---

**1 Input:** a dataset $\mathcal{D}$; reward function $r$. Let $K_0$ denote the number of sample trajectories in $\mathcal{D}$.

**2 Subsampling:** run the following procedure to generate the subsampled dataset $\mathcal{D}^{\mathsf{trim}}$.

  1) *Data splitting.* Split $\mathcal{D}$ into two halves: $\mathcal{D}^{\mathsf{main}}$ (which contains the first $K_0/2$ trajectories), and $\mathcal{D}^{\mathsf{aux}}$ (which contains the remaining $K_0/2$ trajectories); we let $N_h^{\mathsf{main}}(s)$ (resp. $N_h^{\mathsf{aux}}(s)$) denote the number of sample transitions in $\mathcal{D}^{\mathsf{main}}$ (resp. $\mathcal{D}^{\mathsf{aux}}$) that transition from state $s$ at step $h$.

  2) *Lower bounding* $\{N_h^{\mathsf{main}}(s)\}$ *using* $\mathcal{D}^{\mathsf{aux}}$. For each $s \in \mathcal{S}$ and $1 \leq h \leq H$, compute

  $$N_h^{\mathsf{trim}}(s) := \max\left\{N_h^{\mathsf{aux}}(s) - 10\sqrt{N_h^{\mathsf{aux}}(s)\log\frac{HS}{\delta}}, 0\right\}; \tag{50}$$

  3) *Random subsampling.* Let $\mathcal{D}^{\mathsf{main}'}$ be the set of all sample transitions (i.e., the quadruples taking the form $(s,a,h,s')$) from $\mathcal{D}^{\mathsf{main}}$. Subsample $\mathcal{D}^{\mathsf{main}'}$ to obtain $\mathcal{D}^{\mathsf{trim}}$, such that for each $(s,h) \in \mathcal{S} \times [H]$, $\mathcal{D}^{\mathsf{trim}}$ contains $\min\{N_h^{\mathsf{trim}}(s), N_h^{\mathsf{main}}(s)\}$ sample transitions randomly drawn from $\mathcal{D}^{\mathsf{main}'}$. (We shall also let $N_h^{\mathsf{trim}}(s,a)$ denote the number of samples that visits $(s,a,h)$ in $\mathcal{D}^{\mathsf{trim}}$.)

**3 Run VI-LCB:** set $\mathcal{D}_0 = \mathcal{D}^{\mathsf{trim}}$; run Algorithm 7 to compute a policy $\widehat{\pi}$.

---

**Algorithm 7:** Offline value iteration with lower confidence bounds (Li et al., 2022).

---

1 **Input:** dataset $\mathcal{D}_0$; reward function $r$; target success probability $1 - \delta$.

2 **Initialization:** $\widehat{V}_{H+1} = 0$.

3 **for** $h = H, \cdots, 1$ **do**

4     compute the empirical transition kernel $\widehat{P}_h$ as

$$\widehat{P}_h(s' \mid s, a) = \begin{cases} \frac{1}{N_h(s,a)} \sum_{i=1}^{N} \mathbb{1}\big\{(s_i, a_i, h_i, s_i') = (s, a, h, s')\big\}, & \text{if } N_h(s,a) > 0, \\ \frac{1}{S}, & \text{else}, \end{cases} \quad (51)$$

    where $N_h(s,a) := \sum_{i=1}^{N} \mathbb{1}\big\{(s_i, a_i, h_i) = (s, a, h)\big\}$ and
    $N_h(s) := \sum_{i=1}^{N} \mathbb{1}\big\{(s_i, h_i) = (s, h)\big\}$.

5     **for** $s \in \mathcal{S}, a \in \mathcal{A}$ **do**

6         compute the penalty term $b_h(s,a)$ as

7

$$\forall (s, a, h) \in \mathcal{S} \times \mathcal{A} \times [H]: \ b_h(s,a) = \min\left\{ \sqrt{\frac{c_{\mathsf{b}} \log \frac{K}{\delta}}{N_h(s,a)} \mathsf{Var}_{\widehat{P}_h(\cdot | s,a)}(\widehat{V}_{h+1})} + c_{\mathsf{b}} H \frac{\log \frac{K}{\delta}}{N_h(s,a)}, \ H \right\}$$

        for some universal constant $c_{\mathsf{b}} > 0$ (e.g., $c_{\mathsf{b}} = 16$); set
        $\widehat{Q}_h(s,a) = \max\big\{r_h(s,a) + \widehat{P}_{h,s,a}\widehat{V}_{h+1} - b_h(s,a), 0\big\}$.

8     **for** $s \in \mathcal{S}$ **do**

9         set $\widehat{V}_h(s) = \max_a \widehat{Q}_h(s,a)$ and $\widehat{\pi}_h(s) \in \arg\max_a \widehat{Q}_h(s,a)$.

10 **Output:** $\widehat{\pi} = \{\widehat{\pi}_h\}_{1 \le h \le H}$.

---

## E   Analysis of Theorem 1

In this section, we present the proof for our main result in Theorem 1. Throughout the proof, we let $\{\mathcal{G}_h\}_{1 \le h \le H}$ denote a sequence of subsets obeying

$$\max_{1 \le h \le H} \max_{(s,a) \in \mathcal{G}_h} \frac{d_h^{\pi^\star}(s,a)}{d_h^{\mathsf{off}}(s,a)} = C^\star(\sigma) \qquad \text{and} \qquad \frac{1}{H} \sum_{h=1}^{H} \sum_{(s,a) \notin \mathcal{G}_h} d_h^{\pi^\star}(s,a) \le \sigma, \qquad (52)$$

as motivated by Definition 2. As it turns out, if $K^{\mathsf{on}} \ge c_1 \frac{H^3 SA}{\varepsilon^2} \log \frac{K}{\delta}$ for some large enough constant $c_1 > 0$, then the claimed result in Theorem 1 follows immediately from the main theory in Li et al. (2023) developed for pure online exploration. As a result, it sufficies to prove the theorem by replacing Condition (24b) with

$$K^{\mathsf{on}} \ge c_1 \frac{H^4 SA\sigma}{\varepsilon^2} \log \frac{K}{\delta} \qquad (53)$$

throughout this section.

On a high level, our proof comprises the following three steps:

- Establish the proximity of $\widehat{d}^{\mathsf{off}}$ (resp. $\widehat{d}^\pi$) and $d^{\mathsf{off}}$ (resp. $d^\pi$).
- Show that the mixed policy $\pi^{\mathsf{imitate}}$ is able to mimic and strengthen the offline dataset $\mathcal{D}^{\mathsf{off}}$, while the mixed policy $\pi^{\mathsf{explore}}$ is capable of exploring the part of the state-action space that has not been adequately visited by $\mathcal{D}^{\mathsf{off}}$.
- Derive the sub-optimality of the policy returned by the offline RL algorithm (i.e., Algorithm 6) when applied to the hybrid dataset $\mathcal{D} = \mathcal{D}^{\mathsf{off},2} \cup \mathcal{D}^{\mathsf{on}}_{\mathsf{imitate}} \cup \mathcal{D}^{\mathsf{on}}_{\mathsf{explore}}$.

In the sequel, we shall elaborate on these three steps.

### E.1 Step 1: establishing the proximity of $\widehat{d}^\pi$ (resp. $\widehat{d}^{\text{off}}$) and $d^\pi$ (resp. $d^{\text{off}}$)

To begin with, the goodness of the occupancy distribution estimators $\widehat{d}^\pi$ (cf. Algorithm 3) has been analyzed in Li et al. (2023, Lemma 4), which come with the following performance guarantees.

**Lemma 1** (Li et al. (2023)). *Recall that $\xi = c_\xi H^3 S^3 A^3 \log \frac{HSA}{\delta}$ for some large enough constant $c_\xi > 0$. With probability at least $1 - \delta$, the estimated occupancy distributions specified in Algorithm 3 satisfy*

$$\frac{1}{2}\widehat{d}_h^\pi(s,a) - \frac{\xi}{4N} \leq d_h^\pi(s,a) \leq 2\widehat{d}_h^\pi(s,a) + 2e_h^\pi(s,a) + \frac{\xi}{4N} \tag{54}$$

*simultaneously for all $(s,a,h) \in \mathcal{S} \times \mathcal{A} \times [H]$ and all deterministic policy $\pi \in \Pi$, provided that*

$$K^{\text{on}} \geq C_N H^{18} S^{14} A^{14} \log^2 \frac{HSA}{\delta} \tag{55}$$

*for some large enough constant $C_N > 0$. Here, $\{e_h^\pi(s,a)\}$ is some non-negative sequence satisfying*

$$\sum_{s,a} e_h^\pi(s,a) \leq \frac{2SA}{K^{\text{on}}} + \frac{13SAH\xi}{N} \qquad \text{for all } h \in [H] \text{ and all deterministic Markov policy } \pi. \tag{56}$$

We now turn to the estimator $\widehat{d}^{\text{off}}$ (cf. (15)) for the occupancy distribution of the offline dataset, for which we begin with the following lemma concerning the proximity of $d_h^{\text{off}}$ and $\widehat{d}_h^{\text{off}}$. The proof of this lemma is deferred to Section G.1.

**Lemma 2.** *Suppose that $c_{\text{off}} \geq 48$. With probability at least $1 - \delta/3$, one has*

$$\frac{1}{3}\widehat{d}_h^{\text{off}}(s,a) \leq d_h^{\text{off}}(s,a) \leq \widehat{d}_h^{\text{off}}(s,a) + 5c_{\text{off}}\left\{ \frac{\log \frac{HSA}{\delta}}{K^{\text{off}}} + \frac{H^4 S^4 A^4 \log \frac{HSA}{\delta}}{N} + \frac{SA}{K^{\text{on}}} \right\} \tag{57}$$

*simultaneously for all $(s,a,h) \in \mathcal{S} \times \mathcal{A} \times [H]$.*

This lemma implies that: when $d_h^{\text{off}}(s,a) \lesssim \frac{\log \frac{HSA}{\delta}}{K^{\text{off}}} + \frac{H^4 S^4 A^4 \log \frac{HSA}{\delta}}{N} + \frac{SA}{K^{\text{on}}}$, the estimator $\widehat{d}_h^{\text{off}}(s,a)$ might be unable to track $d_h^{\text{off}}(s,a)$ in a faithful manner. This motivates us to single out the following two subsets of state-action pairs for which $\widehat{d}_h^{\text{off}}(s,a)$ might become problematic at step $h$:

- the set $\mathcal{G}_h^c$ (see (11) for the definition of $\mathcal{G}_h$), which corresponds to the set of optimal state-action pairs that even the true data distribution $d_h^{\text{off}}$ cannot cover adequately;

- another set $\mathcal{T}_h^{\text{small}}$ defined as

$$\mathcal{T}_h^{\text{small}} := \left\{ (s,a) : d_h^{\text{off}}(s,a) \leq 10c_{\text{off}}\left( \frac{\log \frac{HSA}{\delta}}{K^{\text{off}}} + \frac{H^4 S^4 A^4 \log \frac{HSA}{\delta}}{N} + \frac{SA}{K^{\text{on}}} \right) \right\}, \tag{58}$$

  comprising those state-action pairs for which $\widehat{d}_h^{\text{off}}(s,a)$ might not be a faithful estimator of $d_h^{\text{off}}(s,a)$.

In what follow, we shall adopt the notation:

$$\mathcal{T}_h := \mathcal{G}_h^c \cup \mathcal{T}_h^{\text{small}}. \tag{59}$$

It is straightforward to demonstrate that:

- For any $(s,a) \notin \mathcal{T}_h^{\text{small}}$, it is seen from Lemma 2 that

$$d_h^{\text{off}}(s,a) \leq \widehat{d}_h^{\text{off}}(s,a) + \frac{1}{2}d_h^{\text{off}}(s,a) \qquad \Longleftrightarrow \qquad d_h^{\text{off}}(s,a) \leq 2\widehat{d}_h^{\text{off}}(s,a). \tag{60}$$

- For any $(s,a) \in \mathcal{G}_h$, Condition (52) tells us that

$$d_h^{\pi^\star}(s,a) \leq C^\star(\sigma)d_h^{\text{off}}(s,a). \tag{61}$$

As a consequence, any $(s, a) \notin \mathcal{T}_h$ necessarily obeys

$$d_h^{\pi^\star}(s, a) \le C^\star(\sigma) d_h^{\mathsf{off}}(s, a) \le 2C^\star(\sigma) \widehat{d}_h^{\mathsf{off}}(s, a). \tag{62}$$

Another useful observation that we can readily make is as follows:

$$
\begin{aligned}
\sum_{h=1}^{H} \sum_{(s,a) \in \mathcal{T}_h} d_h^{\pi^\star}(s, a) &\le \sum_{h=1}^{H} \sum_{(s,a) \notin \mathcal{G}_h} d_h^{\pi^\star}(s, a) + \sum_{h=1}^{H} \sum_{(s,a) \in \mathcal{G}_h \cap \mathcal{T}_h^{\mathsf{small}}} d_h^{\pi^\star}(s, a) \\
&\le H\sigma + \sum_{h=1}^{H} \sum_{(s,a) \in \mathcal{G}_h \cap \mathcal{T}_h^{\mathsf{small}}} d_h^{\pi^\star}(s, a) \\
&\le H\sigma + C^\star(\sigma) \sum_{h=1}^{H} \sum_{(s,a) \in \mathcal{T}_h^{\mathsf{small}}} d_h^{\mathsf{off}}(s, a) \mathbb{1}\left(a = \pi^\star(s)\right) \\
&\le H\sigma + C^\star(\sigma) HS \cdot 10 c_{\mathsf{off}} \left( \frac{\log \frac{HSA}{\delta}}{K^{\mathsf{off}}} + \frac{H^4 S^4 A^4 \log \frac{HSA}{\delta}}{N} + \frac{SA}{K^{\mathsf{on}}} \right) \\
&\le H\sigma + 10 c_{\mathsf{off}} \left( \frac{C^\star(\sigma) HS \log \frac{HSA}{\delta}}{K^{\mathsf{off}}} + \frac{4 C^\star(\sigma) H^6 S^5 A^4 \log \frac{HSA}{\delta}}{K^{\mathsf{on}}} \right) =: \widehat{\sigma}.
\end{aligned}
\tag{63}
$$

Here, the second and the third lines arise from Condition (52), the penultimate line invokes the definition (58) of $\mathcal{T}_h^{\mathsf{small}}$, whereas the last line is valid since $N = K^{\mathsf{on}}/(3H)$ (see (14)).

## E.2  Step 2: showing that $\pi^{\mathsf{imitate}}$ (resp. $\pi^{\mathsf{explore}}$) covers $\widehat{d}^{\mathsf{off}}$ (resp. $d^{\pi^\star}$) adequately

In this step, we aim to demonstrate the quality of the two exploration policies $\pi^{\mathsf{imitate}}$ and $\pi^{\mathsf{explore}}$, designed for different purposes.

**Goodness of $\pi^{\mathsf{imitate}}$.** We begin by assessing the quality of the exploration policy $\pi^{\mathsf{imitate}}$. Towards this, we first make note of the following crude bound:

$$\frac{\widehat{d}_h^{\mathsf{off}}(s, a)}{\frac{1}{K^{\mathsf{on}} H} + \mathbb{E}_{\pi' \sim \mu^t}\left[\widehat{d}_h^\pi(s, a)\right]} \le \frac{\widehat{d}_h^{\mathsf{off}}(s, a)}{\frac{1}{K^{\mathsf{on}} H}} \le K^{\mathsf{on}} H =: L.$$

In view of the convergence guarantees for FTRL (Shalev-Shwartz, 2012, Corollary 2.16), we see that: if $\eta = \sqrt{\frac{\log A}{2 T_{\mathsf{max}} L^2}} = \sqrt{\frac{\log A}{2 T_{\mathsf{max}} (K^{\mathsf{on}} H)^2}}$, then running FTRL for $T_{\mathsf{max}}$ iterations results in

$$
\begin{aligned}
\max_{a \in \mathcal{A}} \frac{1}{T_{\mathsf{max}}} \sum_{t=1}^{T_{\mathsf{max}}} \frac{\widehat{d}_h^{\mathsf{off}}(s, a)}{\frac{1}{K^{\mathsf{on}} H} + \mathbb{E}_{\pi \sim \mu^t}\left[\widehat{d}_h^\pi(s, a)\right]} &- \frac{1}{T_{\mathsf{max}}} \sum_{t=1}^{T_{\mathsf{max}}} \sum_{a \in \mathcal{A}} \pi_h^t(a \mid s) \frac{\widehat{d}_h^{\mathsf{off}}(s, a)}{\frac{1}{K^{\mathsf{on}} H} + \mathbb{E}_{\pi \sim \mu^t}\left[\widehat{d}_h^\pi(s, a)\right]} \\
&\le K^{\mathsf{on}} H \sqrt{\frac{2 \log A}{T_{\mathsf{max}}}}
\end{aligned}
\tag{64}
$$

for all $s \in \mathcal{S}$ and $1 \le h \le H$. Therefore, recalling that $\mu^{\mathsf{imitate}} = \frac{1}{T_{\mathsf{max}}} \sum_{t=1}^{T_{\mathsf{max}}} \mu^t$ and applying Jensen's inequality yield

$$
\begin{aligned}
\sum_{h=1}^{H} \sum_{s \in \mathcal{S}} \max_{a \in \mathcal{A}} & \frac{\widehat{d}_h^{\mathsf{off}}(s, a)}{\frac{1}{K^{\mathsf{on}} H} + \mathbb{E}_{\pi \sim \mu^{\mathsf{imitate}}}\left[\widehat{d}_h^\pi(s, a)\right]} \\
&\le \sum_{h=1}^{H} \sum_{s \in \mathcal{S}} \max_{a \in \mathcal{A}} \frac{1}{T_{\mathsf{max}}} \sum_{t=1}^{T_{\mathsf{max}}} \frac{\widehat{d}_h^{\mathsf{off}}(s, a)}{\frac{1}{K^{\mathsf{on}} H} + \mathbb{E}_{\pi \sim \mu^t}\left[\widehat{d}_h^\pi(s, a)\right]} \\
&\le \sum_{h=1}^{H} \sum_{(s,a) \in \mathcal{S} \times \mathcal{A}} \frac{1}{T_{\mathsf{max}}} \sum_{t=1}^{T_{\mathsf{max}}} \pi_h^t(a \mid s) \frac{\widehat{d}_h^{\mathsf{off}}(s, a)}{\frac{1}{K^{\mathsf{on}} H} + \mathbb{E}_{\pi \sim \mu^t}\left[\widehat{d}_h^\pi(s, a)\right]} + K^{\mathsf{on}} H^2 S \sqrt{\frac{2 \log A}{T_{\mathsf{max}}}}, \quad (65)
\end{aligned}
$$

where the second inequality results from (64). In addition, it follows from the stopping rule (37) that

$$\sum_{h=1}^{H} \sum_{(s,a)\in\mathcal{S}\times\mathcal{A}} \pi_h^t(a\,|\,s) \frac{\widehat{d}_h^{\mathsf{off}}(s,a)}{\frac{1}{K^{\mathsf{on}}H} + \mathbb{E}_{\pi\sim\mu^t}\big[\widehat{d}_h^{\pi}(s,a)\big]} \le 108SH. \tag{66}$$

As a consequence, combining (65) and (66) yields

$$\sum_{h\in[H]} \sum_{s\in\mathcal{S}} \max_{a\in\mathcal{A}} \frac{\widehat{d}_h^{\mathsf{off}}(s,a)}{\frac{1}{K^{\mathsf{on}}H} + \mathbb{E}_{\pi\sim\mu^{\mathsf{imitate}}}\big[\widehat{d}_h^{\pi}(s,a)\big]} \le 108SH + K^{\mathsf{on}}H^2 S\sqrt{\frac{2\log A}{T_{\mathsf{max}}}} \le 109SH, \tag{67}$$

provided that $T_{\mathsf{max}} \ge 2(K^{\mathsf{on}}H)^2 \log A$. The fact that the left-hand side of (67) is well-controlled suggests that $\pi^{\mathsf{imitate}}$ is able to cover $\widehat{d}^{\mathsf{off}}$ adequately, a crucial fact we shall rely on in the subsequent analysis.

**Goodness of $\pi^{\mathsf{explore}}$.** Next, we turn attention to the other exploration policy $\pi^{\mathsf{explore}}$, computed via Algorithm 5. The following performance guarantees have been established in Li et al. (2023, Section 3.2).

**Lemma 3.** *The distribution $\mu^{\mathsf{explore}} \in \Delta(\Pi)$ returned by Algorithm 5 satisfies*

$$\max_{\pi} \sum_{h=1}^{H} \sum_{(s,a)\in\mathcal{S}\times\mathcal{A}} \frac{\widehat{d}_h^{\pi}(s,a)}{\frac{1}{K^{\mathsf{on}}H} + \mathbb{E}_{\pi'\sim\mu^{\mathsf{explore}}}\big[\widehat{d}_h^{\pi'}(s,a)\big]} \le 2HSA.$$

In light of the performance bound (17) for the subsequent offline RL approach, Lemma 3 suggests that $\pi^{\mathsf{explore}}$ is able to explore well with regards to the visitation of any policy $\pi$ — including the optimal policy $\pi^{\star}$.

### E.3 Step 3: establishing the performance of offline RL

Now, we can readily proceed to analyze the performance of the model-based offline procedure described in Algorithm 6. In this subsection, we abuse the notation $\widehat{P}$ to represent the empirical transition kernel constructed within the offline subroutine in Algorithm 7. Additionally, we introduce a $S$-dimensional vector $d_h^{\pi^\star} := [d_h^{\pi^\star}(s)]_{s\in\mathcal{S}}$.

#### E.3.1 Step 3.1: error decomposition

To begin with, we convert the sub-optimality gap of the policy estimate $\widehat{\pi}$ into several terms that shall be controlled separately. The following two preliminary facts, which have been established in Li et al. (2022), prove useful for this purpose.

**Lemma 4.** *With probability exceeding $1 - \delta/3$, one has*

$$N_h^{\mathsf{main}}(s,a) \ge N_h^{\mathsf{trim}}(s,a), \qquad \forall(s,a,h)\in\mathcal{S}\times\mathcal{A}\times[H]$$

*and*

$$\big\langle d_j^{\pi^\star}, V_j^\star - V_j^{\widehat{\pi}} \big\rangle \le 2 \sum_{h:h\ge j} \sum_{s,a} d_h^{\pi^\star}(s,a) b_h(s,a), \qquad 1\le h\le H,$$

*where $b_h(s,a)$ is defined in line 7 of Algorithm 7.*

In view of Lemma 4, we can derive, for all $j\in[H]$,

$$\big\langle d_j^{\pi^\star}, V_j^\star - V_j^{\widehat{\pi}} \big\rangle \le 2 \sum_{h:h\ge j} \sum_{s,a} d_h^{\pi^\star}(s,a) b_h(s,a) = 2 \sum_{h:h\ge j} \sum_{s} d_h^{\pi^\star}\big(s,\pi_h^\star(s)\big) b_h\big(s,\pi_h^\star(s)\big)$$

$$\le 2 \sum_{h:h\ge j} \sum_{s\,:\,(s,\pi^\star(s))\notin\mathcal{T}_h} \sqrt{2d_h^{\pi^\star}\big(s,\pi_h^\star(s)\big) C^\star(\sigma)\widehat{d}_h^{\mathsf{off}}\big(s,\pi_h^\star(s)\big)} \, b_h\big(s,\pi_h^\star(s)\big) + 2 \sum_{h:h\ge j} \sum_{(s,a)\in\mathcal{T}_h} d_h^{\pi^\star}(s,a) b_h(s,a)$$

$$\le 2 \sum_{h:h\ge j} \sum_{s\,:\,(s,\pi^\star(s))\notin\mathcal{T}_h} \sqrt{2d_h^{\pi^\star}\big(s,\pi_h^\star(s)\big) C^\star(\sigma)\widehat{d}_h^{\mathsf{off}}\big(s,\pi_h^\star(s)\big)} \, b_h\big(s,\pi_h^\star(s)\big)$$

$$+ 4 \sum_{h:h\geq j} \sum_{(s,a)\in\mathcal{T}_h} \widehat{d}_h^{\pi^\star}(s,a) b_h(s,a) + \frac{8H^2 SA}{K^{\text{on}}} + \frac{53c_\xi H^6 S^4 A^4}{N} \log\frac{HSA}{\delta}.$$

Here, the second line comes from (62), whereas the third line is due to Lemma 1 and the basic fact that $b_h(s,a) \leq H$ (see line 7 of Algorithm 7). Substituting the definition of $b_h$ (see line 7 of Algorithm 7) into the above display and applying Lemma 4, we arrive at

$$\left\langle d_j^{\pi^\star}, V_j^\star - V_j^{\widehat{\pi}} \right\rangle \leq \sum_{h:h\geq j} \sum_s \max_{a:(s,a)\notin\mathcal{T}_h} \left\{ \sqrt{8 d_h^{\pi^\star}(s,a) C^\star(\sigma) \widehat{d}_h^{\text{off}}(s,a)} \cdot \right.$$
$$\left. \min\left\{ \sqrt{\frac{c_{\mathsf{b}} \log\frac{K}{\delta}}{N_h^{\text{trim}}(s,a)} \mathsf{Var}_{\widehat{P}_h(\cdot|s,a)}(\widehat{V}_{h+1})} + \frac{c_{\mathsf{b}} H \log\frac{K}{\delta}}{N_h^{\text{trim}}(s,a)}, H \right\} \right\}$$
$$+ 4H \sum_{h:h\geq j} \sum_{(s,a)\in\mathcal{T}_h} \widehat{d}_h^{\pi^\star}(s,a) \sqrt{\frac{c_{\mathsf{b}} \log\frac{K}{\delta}}{N_h^{\text{trim}}(s,a)+1}} + \frac{8H^2 SA}{K^{\text{on}}} + \frac{53c_\xi H^6 S^4 A^4}{N} \log\frac{K}{\delta}, \tag{68}$$

where we recall that $c_{\mathsf{b}} > 0$ is also an absolute constant used to specify $b_h(s,a)$.

It is worth noting that the right-hand side of (68) involves a variance term $\mathsf{Var}_{\widehat{P}_h(\cdot|s,a)}(\widehat{V}_{h+1})$ w.r.t. the empirical model $\widehat{P}$. As it turns out, the following lemma established in Li et al. (2022, Lemma 8) makes apparent the intimate connection between $\mathsf{Var}_{\widehat{P}_h(\cdot|s,a)}(\widehat{V}_{h+1})$ and $\mathsf{Var}_{P_h(\cdot|s,a)}(\widehat{V}_{h+1})$.

**Lemma 5.** *With probability exceeding $1 - \delta/3$, we have, for all $(s,a,h) \in \mathcal{S} \times \mathcal{A} \times [H]$,*

$$\mathsf{Var}_{\widehat{P}_h(\cdot|s,a)}(\widehat{V}_{h+1}) \leq 2\mathsf{Var}_{P_h(\cdot|s,a)}(\widehat{V}_{h+1}) + \frac{10H^2 \log\frac{K}{\delta}}{3N_h^{\text{trim}}(s,a)}.$$

Substituting the result of Lemma 5 into (68) leads to

$$\left\langle d_j^{\pi^\star}, V_j^\star - V_j^{\widehat{\pi}} \right\rangle \leq \gamma_1 + \gamma_2 + \frac{8H^2 SA}{K^{\text{on}}} + \frac{53c_\xi H^6 S^4 A^4}{N} \log\frac{K}{\delta}, \tag{69}$$

where

$$\gamma_1 := \sum_{h:h\geq j} \sum_s \max_{a:(s,a)\notin\mathcal{T}_h} \left\{ 2\sqrt{2 d_h^{\pi^\star}(s,a) C^\star(\sigma) \widehat{d}_h^{\text{off}}(s,a)} \min\left\{ \sqrt{\frac{2c_{\mathsf{b}} \log\frac{K}{\delta}}{N_h^{\text{trim}}(s,a)} \mathsf{Var}_{P_h(\cdot|s,a)}(\widehat{V}_{h+1})} + \frac{4c_{\mathsf{b}} H \log\frac{K}{\delta}}{N_h^{\text{trim}}(s,a)}, H \right\} \right\};$$

$$\gamma_2 := 4H \sum_{h:h\geq j} \sum_{(s,a)\in\mathcal{T}_h} \widehat{d}_h^{\pi^\star}(s,a) \sqrt{\frac{c_{\mathsf{b}} \log\frac{K}{\delta}}{N_h^{\text{trim}}(s,a)+1}}.$$

This leaves us with two terms to bound, which we shall accomplish separately in the ensuing two steps.

### E.3.2 Step 3.2: controlling $\gamma_1$ in (69)

Regarding the first term $\gamma_1$ on the right-hand side of (69), let us first define the set $\mathcal{I}_h$ as follows:

$$\mathcal{I}_h := \left\{ (s,a) : \mathbb{E}_{\pi\sim\mu^{\text{imitate}}}\left[\widehat{d}_h^\pi(s,a)\right] \geq \frac{\xi}{N} \right\}, \tag{70}$$

where we remind the reader that $\xi = c_\xi H^3 S^3 A^3 \log\frac{HSA}{\delta}$ for some constant $c_\xi > 0$. Armed with this set, we can deduce that

$$\sum_{h:h\geq j} \sum_s \max_{a:(s,a)\notin\mathcal{I}_h\cup\mathcal{T}_h} \sqrt{2 d_h^{\pi^\star}(s,a) C^\star(\sigma) \widehat{d}_h^{\text{off}}(s,a)}$$
$$\leq \sum_{h:h\geq j} \sum_s 2 \max_{a:(s,a)\notin\mathcal{I}_h} C^\star(\sigma) \widehat{d}_h^{\text{off}}(s,a)$$

$$\leq 2C^{\star}(\sigma)\Big(\frac{1}{K^{\mathsf{on}}H} + \frac{\xi}{N}\Big) \sum_{h:h\geq j} \sum_s \max_a \frac{\widehat{d}_h^{\mathsf{off}}(s,a)}{\frac{1}{K^{\mathsf{on}}H} + \mathbb{E}_{\pi\sim\mu^{\mathsf{imitate}}}\big[\widehat{d}_h^{\pi}(s,a)\big]}$$

$$\leq 218HSC^{\star}(\sigma)\Big(\frac{\xi}{N} + \frac{1}{K^{\mathsf{on}}H}\Big),$$

where the first inequality arises from (62), the penulminate line utilizes the definition (70) of $\mathcal{I}_h$, and the last line comes from (67). This in turn allows us to upper bound $\gamma_1$ as follows:

$$\gamma_1 \leq \sum_{h:h\geq j} \sum_s 2 \max_{a:(s,a)\in\mathcal{I}_h} \sqrt{2d_h^{\pi^{\star}}(s,a)C^{\star}(\sigma)\widehat{d}_h^{\mathsf{off}}(s,a)} \min\Bigg\{\sqrt{\frac{2c_{\mathsf{b}}\log\frac{HK}{\delta}}{N_h^{\mathsf{trim}}(s,a)}\mathsf{Var}_{P_h(\cdot|s,a)}\big(\widehat{V}_{h+1}\big)} + \frac{4c_{\mathsf{b}}H\log\frac{K}{\delta}}{N_h^{\mathsf{trim}}(s,a)}, H\Bigg\}$$

$$+ \sum_{h:h\geq j} \sum_s 2 \max_{a:(s,a)\notin\mathcal{I}_h\cup\mathcal{T}_h} \sqrt{2d_h^{\pi^{\star}}(s,a)C^{\star}(\sigma)\widehat{d}_h^{\mathsf{off}}(s,a)} \cdot H$$

$$\leq \sum_{h:h\geq j} \sum_s 2 \max_{a:(s,a)\in\mathcal{I}_h} \sqrt{2d_h^{\pi^{\star}}(s,a)C^{\star}(\sigma)\widehat{d}_h^{\mathsf{off}}(s,a)} \min\Bigg\{\sqrt{\frac{2c_{\mathsf{b}}\log\frac{HK}{\delta}}{N_h^{\mathsf{trim}}(s,a)}\mathsf{Var}_{P_h(\cdot|s,a)}\big(\widehat{V}_{h+1}\big)} + \frac{4c_{\mathsf{b}}H\log\frac{K}{\delta}}{N_h^{\mathsf{trim}}(s,a)}, H\Bigg\}$$

$$+ 436H^2SC^{\star}(\sigma)\Big(\frac{\xi}{N} + \frac{1}{K^{\mathsf{on}}H}\Big)$$

$$\leq 16c_{\mathsf{b}} \sum_{h:h\geq j} \sum_s \max_{a:(s,a)\in\mathcal{I}_h} \sqrt{2d_h^{\pi^{\star}}(s,a)C^{\star}(\sigma)\widehat{d}_h^{\mathsf{off}}(s,a)} \sqrt{\frac{\mathsf{Var}_{P_h(\cdot|s,a)}\big(\widehat{V}_{h+1}\big) + H}{N_h^{\mathsf{trim}}(s,a) + 1/H}\log^2\frac{K}{\delta}}$$

$$+ 436H^2SC^{\star}(\sigma)\Big(\frac{\xi}{N} + \frac{1}{K^{\mathsf{on}}H}\Big), \tag{71}$$

where the last line makes use of the elementary fact that $\min\big\{\frac{x}{y}, \frac{u}{w}\big\} \leq \frac{x+u}{y+w}$ for any $x,y,u,w > 0$.

In addition, note that for any $s$ obeying $\mathbb{E}_{\pi\sim\mu^{\mathsf{imitate}}}\big[d_h^{\pi}(s)\big] \geq \xi/N$, we have

$$\mathbb{E}\big[N_h^{\mathsf{aux}}(s)\big] = \frac{1}{4}K^{\mathsf{off}}d_h^{\mathsf{off}}(s) + \frac{1}{6}K^{\mathsf{on}}\mathbb{E}_{\pi\sim\mu^{\mathsf{imitate}}}\big[d_h^{\pi}(s)\big] + \frac{1}{6}K^{\mathsf{on}}\mathbb{E}_{\pi\sim\mu^{\mathsf{explore}}}\big[d_h^{\pi}(s)\big]$$

$$\geq \frac{1}{6}K^{\mathsf{on}}\mathbb{E}_{\pi\sim\mu^{\mathsf{imitate}}}\big[d_h^{\pi}(s)\big] \geq \frac{1}{6}K^{\mathsf{on}} \cdot \frac{\xi}{N} = \frac{1}{2}c_{\xi}H^4S^3A^3\log\frac{HSA}{\delta},$$

where the last line invokes the definition of $\mathcal{I}_h$ and the choice $NH = \frac{1}{3}K^{\mathsf{on}}$. It can then be straight-forwardly justified using elementary concentration inequalities (see, e.g., Alon and Spencer (2016, Appendix A.1)) that: with probability exceeding $1 - \delta/10$,

$$N_h^{\mathsf{aux}}(s) \geq \frac{1}{2}\mathbb{E}\big[N_h^{\mathsf{aux}}(s)\big] \geq \frac{1}{4}c_{\xi}H^4S^3A^3\log\frac{HSA}{\delta}$$

holds simultaneously for all $(s,h) \in \mathcal{S} \times [H]$, and as a result,

$$N_h^{\mathsf{trim}}(s) \geq N_h^{\mathsf{aux}}(s) - 10\sqrt{N_h^{\mathsf{aux}}(s)\log\frac{HS}{\delta}} \geq \frac{1}{2}N_h^{\mathsf{aux}}(s) \geq \frac{1}{4}\mathbb{E}\big[N_h^{\mathsf{aux}}(s)\big] \geq \frac{1}{24}K^{\mathsf{on}}\mathbb{E}_{\pi\sim\mu^{\mathsf{imitate}}}\big[d_h^{\pi}(s)\big].$$

Moreover, for any $(s,a) \in \mathcal{I}_h$ (cf. (70)), one can invoke Lemma 2 to obtain

$$\mathbb{E}_{\pi\sim\mu^{\mathsf{imitate}}}\big[d_h^{\pi}(s)\big] \geq \frac{1}{3}\mathbb{E}_{\pi\sim\mu^{\mathsf{imitate}}}\big[\widehat{d}_h^{\pi}(s)\big] \geq \frac{\xi}{3N}.$$

Applying the same concentration of measurement argument as above further reveals that:

$$N_h^{\mathsf{trim}}(s,a) \geq \frac{1}{24}K^{\mathsf{on}}\mathbb{E}_{\pi\sim\mu^{\mathsf{imitate}}}\big[d_h^{\pi}(s,a)\big] \geq \frac{1}{72}K^{\mathsf{on}}\mathbb{E}_{\pi\sim\mu^{\mathsf{imitate}}}\big[\widehat{d}_h^{\pi}(s,a)\big]$$

any $(s,a) \in \mathcal{I}_h$. Substitution into (71) then gives

$$\gamma_1 \leq 16c_{\mathsf{b}} \sum_{h:h\geq j} \sum_s \max_{a:(s,a)\in\mathcal{I}_h} \sqrt{2d_h^{\pi^{\star}}(s,a)C^{\star}(\sigma)\widehat{d}_h^{\mathsf{off}}(s,a)} \sqrt{\frac{\mathsf{Var}_{P_h(\cdot|s,a)}\big(\widehat{V}_{h+1}\big) + H}{1/H + \frac{1}{72}K^{\mathsf{on}}\mathbb{E}_{\pi\sim\mu^{\mathsf{imitate}}}\big[\widehat{d}_h^{\pi}(s,a)\big]}\log^2\frac{K}{\delta}}$$

$$+ 436H^2 SC^\star(\sigma)\Big(\frac{\xi}{N} + \frac{1}{K^{\text{on}}H}\Big). \tag{72}$$

By virtue of the Cauchy-Schwarz inequality, we can further derive

$$\sum_{h:h\geq j} \sum_s \max_a \sqrt{d_h^{\pi^\star}(s,a)C^\star(\sigma)\widehat{d}_h^{\text{off}}(s,a)}\sqrt{\frac{\mathsf{Var}_{P_h(\cdot|s,a)}\big(\widehat{V}_{h+1}\big) + H}{1/H + \frac{1}{72}K^{\text{on}}\mathbb{E}_{\pi\sim\mu^{\text{imitate}}}\big[\widehat{d}_h^\pi(s,a)\big]}}$$

$$\leq \sqrt{\sum_{h:h\geq j} \sum_{s,a} d_h^{\pi^\star}(s,a)\Big(\mathsf{Var}_{P_h(\cdot|s,a)}\big(\widehat{V}_{h+1}\big) + H\Big)} \cdot \sqrt{\sum_{h:h\geq j} \sum_s \max_a \frac{C^\star(\sigma)\widehat{d}_h^{\text{off}}(s,a)}{1/H + \frac{1}{72}K^{\text{on}}\mathbb{E}_{\pi\sim\mu^{\text{imitate}}}\big[\widehat{d}_h^\pi(s,a)\big]}}. \tag{73}$$

To further control this term, we resort to the following lemma, whose proof is deferred to Section G.2.

**Lemma 6.** *With probability at least $1 - \delta/6$, we have, for all $j \in [H]$,*

$$\sum_{h:h\geq j} \sum_{s,a} d_h^{\pi^\star}(s,a)\mathsf{Var}_{P_h(\cdot|s,a)}\big(\widehat{V}_{h+1}\big) \leq 5H^2,$$

*provided that*

$$K^{\text{on}} \geq c_{11}\big(H^{18}S^{14}A^{14} + H^5 S^4 A^3 C^\star(\sigma)\big)\log^2 \frac{K}{\delta}$$

$$K^{\text{off}} \geq c_{11} HS\big(C^\star(\sigma) + A\big)\log \frac{K}{\delta}$$

*for some sufficiently large constant $c_{11} > 0$.*

Putting Lemma 6 together with (67) and (73), we obtain

$$\sum_{h:h\geq j} \sum_s \max_a \sqrt{d_h^{\pi^\star}(s,a)C^\star(\sigma)\widehat{d}_h^{\text{off}}(s,a)}\sqrt{\frac{\mathsf{Var}_{P_h(\cdot|s,a)}\big(\widehat{V}_{h+1}\big) + H}{1/H + \frac{1}{72}K^{\text{on}}\mathbb{E}_{\pi\in\mu^{\text{imitate}}}\big[\widehat{d}_h^\pi(s,a)\big]}} \lesssim \sqrt{\frac{H^3 SC^\star(\sigma)}{K^{\text{on}}}}. \tag{74}$$

Substitution into (72) results in

$$\gamma_1 \leq \sqrt{\frac{H^3 SC^\star(\sigma)\log^2 \frac{K}{\delta}}{K^{\text{on}}}} + H^2 SC^\star(\sigma)\Big(\frac{\xi}{N} + \frac{1}{K^{\text{on}}H}\Big). \tag{75}$$

Akin to (72) and (75), we can also focus on the offline dataset and obtain

$$\gamma_1 \lesssim \sqrt{\frac{H^3 SC^\star(\sigma)}{K^{\text{off}}}\log^2 \frac{K}{\delta}} + H^2 SC^\star(\sigma)\Big(\frac{\xi}{N} + \frac{1}{K^{\text{off}}H}\Big). \tag{76}$$

Combine (75) and (76) to arrive at

$$\gamma_1 \lesssim \min\left\{\sqrt{\frac{H^3 SC^\star(\sigma)}{K^{\text{on}}}\log^2 \frac{K}{\delta}}, \sqrt{\frac{H^3 SC^\star(\sigma)}{K^{\text{off}}}\log^2 \frac{K}{\delta}}\right\} + H^2 SC^\star(\sigma)\Big(\frac{\xi}{N} + \frac{1}{\min\{K^{\text{on}}, K^{\text{off}}\}H}\Big)$$

$$\lesssim \sqrt{\frac{H^3 SC^\star(\sigma)}{\max\{K^{\text{on}}, K^{\text{off}}\}}\log^2 \frac{K}{\delta}} + H^2 SC^\star(\sigma)\Big(\frac{\xi}{N} + \frac{1}{\min\{K^{\text{on}}, K^{\text{off}}\}H}\Big)$$

$$\lesssim \sqrt{\frac{H^3 SC^\star(\sigma)}{K^{\text{on}} + K^{\text{off}}}\log^2 \frac{K}{\delta}} + H^2 SC^\star(\sigma)\Big(\frac{\xi}{N} + \frac{1}{\min\{K^{\text{on}}, K^{\text{off}}\}H}\Big). \tag{77}$$

### E.3.3 Step 3.3: controlling $\gamma_2$ in (69)

We now turn attention to the term $\gamma_2$ on the right-hand side of (69). Akin to (72), we can deduce that

$$\gamma_2 \leq 16c_{\mathsf{b}}H \sum_{h:h\geq j} \sum_{(s,a)\in\mathcal{T}_h} \widehat{d}_h^{\pi^\star}(s,a)\sqrt{\frac{\log \frac{K}{\delta}}{1 + \frac{1}{72}K^{\text{on}}\mathbb{E}_{\pi\sim\mu^{\text{explore}}}\big[\widehat{d}_h^\pi(s,a)\big]}} + 436H^2 SA\Big(\frac{\xi}{N} + \frac{1}{K^{\text{on}}H}\Big). \tag{78}$$

The Cauchy-Schwarz inequality then tells us that

$$
\sum_{h:h\geq j}\sum_{(s,a)\in\mathcal{T}_h}\widehat{d}_h^{\pi^\star}(s,a)\sqrt{\frac{1}{1+\frac{1}{72}K^{\mathsf{on}}\mathbb{E}_{\pi\sim\mu^{\mathsf{explore}}}\big[\widehat{d}_h^\pi(s,a)\big]}}
$$

$$
\leq \sqrt{\sum_{h:h\geq j}\sum_{(s,a)\in\mathcal{T}_h}\frac{\widehat{d}_h^{\pi^\star}(s,a)}{1+\frac{1}{72}K^{\mathsf{on}}\mathbb{E}_{\pi\sim\mu^{\mathsf{explore}}}\big[\widehat{d}_h^\pi(s,a)\big]}}\cdot\sqrt{\sum_{h:h\geq j}\sum_{(s,a)\in\mathcal{T}_h}\widehat{d}_h^{\pi^\star}(s,a)}
$$

$$
\leq 6\sqrt{\frac{2HSA}{K^{\mathsf{on}}}}\cdot\sqrt{\sum_{h:h\geq j}\sum_{(s,a)\in\mathcal{T}_h}\widehat{d}_h^{\pi^\star}(s,a)}
$$

$$
\leq 6\sqrt{\frac{2HSA(2\widehat{\sigma}+HS\xi/N)}{K^{\mathsf{on}}}},
$$

where $\widehat{\sigma}$ is defined in (63). Here, the penultimate line invokes Lemma 3, and the last line is valid since (according to Lemma 1 and (63))

$$
\sum_{h:h\geq j}\sum_{(s,a)\in\mathcal{T}_h}\widehat{d}_h^{\pi^\star}(s,a) = \sum_{h:h\geq j}\sum_{s:(s,\pi^\star(s))\in\mathcal{T}_h}\widehat{d}_h^{\pi^\star}\big(s,\pi^\star(s)\big)
$$

$$
\leq 2\sum_{h:h\geq j}\sum_{s:(s,\pi^\star(s))\in\mathcal{T}_h}d_h^{\pi^\star}\big(s,\pi^\star(s)\big)+\frac{HS\xi}{N}
$$

$$
\leq 2\widehat{\sigma}+\frac{HS\xi}{N}.
$$

Substitution of the above inequality into (78) yields

$$
\gamma_2 \leq 96c_{\mathsf{b}}\sqrt{\frac{2H^3SA(2\widehat{\sigma}+HS\xi/N)}{K^{\mathsf{on}}}\log\frac{HK}{\delta}}+436H^2SA\Big(\frac{\xi}{N}+\frac{1}{K^{\mathsf{on}}H}\Big). \qquad (79)
$$

### E.3.4  Step 3.4: putting all pieces together

To finish up, combining (69),(77) and (79) reveals that: with probability at least $1-\delta$, one has

$$
V_1^\star(\rho)-V^{\widehat{\pi}}(\rho) = \big\langle d_1^{\pi^\star}, V_1^\star-V^{\widehat{\pi}}\big\rangle
$$

$$
\lesssim \sqrt{\frac{H^3SC^\star(\sigma)\log^2\frac{K}{\delta}}{K^{\mathsf{on}}+K^{\mathsf{off}}}}+\sqrt{\frac{H^4SA\sigma\log\frac{K}{\delta}}{K^{\mathsf{on}}}}+\sqrt{\frac{H^4S^2AC^\star(\sigma)\log^2\frac{K}{\delta}}{K^{\mathsf{off}}K^{\mathsf{on}}}}
$$

$$
+\sqrt{\frac{H^8S^6A^5C^\star(\sigma)\log^2\frac{K}{\delta}}{NK^{\mathsf{on}}}}+\sqrt{\frac{H^4S^3A^2C^\star(\sigma)\log\frac{K}{\delta}}{KK^{\mathsf{on}}}}
$$

$$
+\frac{H^6S^4A^4+H^5S^4A^3C^\star(\sigma)}{N}\log\frac{K}{\delta}+\frac{H^2S(C^\star(\sigma)+A)}{\min\{K^{\mathsf{on}},K^{\mathsf{off}}\}}
$$

$$
\lesssim \sqrt{\frac{H^3SC^\star(\sigma)\log^2\frac{K}{\delta}}{K^{\mathsf{on}}+K^{\mathsf{off}}}}+\sqrt{\frac{H^4SA\sigma\log\frac{K}{\delta}}{K^{\mathsf{on}}}}+\frac{H^6S^4A^4+H^5S^4A^3C^\star(\sigma)}{K^{\mathsf{on}}}\log^2\frac{K}{\delta}+\frac{H^2S(C^\star(\sigma)+A)}{K^{\mathsf{off}}},
$$
$$(80)$$

where the last inequality holds true as long as $\min\{K^{\mathsf{off}},K^{\mathsf{on}}\}\gtrsim HSA$. Taking the right-hand side of (80) to be no larger than $\varepsilon$, we immediately establish Theorem 1 under the sample complexity assumption in this theorem.

# F Proof for the stopping criterion and the iteration complexity for solving (20b)

**Feasibility of the stopping rule (37).** We first demonstrate that the stopping rule (37) can be satisfied by some mixed policy, namely,

$$\min_{\mu \in \Delta(\Pi)} \sum_{h=1}^{H} \sum_{s \in \mathcal{S}} \mathbb{E}_{a \sim \pi_h^{t+1}(\cdot | s)} \left[ \frac{\widehat{d}_h^{\mathsf{off}}(s,a)}{\frac{1}{KH} + \mathbb{E}_{\pi \sim \mu}\left[\widehat{d}_h^{\pi}(s,a)\right]} \right] \leq 108 SH. \tag{81}$$

Towards this end, we focus attention on analyzing a specific choice of the mixed policy $\mu^{\mathsf{off}}$ — the one that represents the mixed policy that generates the offline dataset. Making use of the definition (15) of $\widehat{d}^{\mathsf{off}}$ gives

$$\widehat{d}_h^{\mathsf{off}}(s,a) = \frac{2N_h^{\mathsf{off}}(s,a)}{K^{\mathsf{off}}} \mathbb{1}\left(\frac{N_h^{\mathsf{off}}(s,a)}{K^{\mathsf{off}}} \geq c_{\mathsf{off}}\left\{\frac{\log \frac{HSA}{\delta}}{K^{\mathsf{off}}} + \frac{H^4 S^4 A^4 \log \frac{HSA}{\delta}}{N} + \frac{SA}{K}\right\}\right)$$

$$\leq 3 d_h^{\mathsf{off}}(s,a) \mathbb{1}\left(\frac{3}{2} d_h^{\mathsf{off}}(s,a) \geq c_{\mathsf{off}}\left\{\frac{\log \frac{HSA}{\delta}}{K^{\mathsf{off}}} + \frac{H^4 S^4 A^4 \log \frac{HSA}{\delta}}{N} + \frac{SA}{K}\right\}\right)$$

$$\leq 3 d_h^{\mathsf{off}}(s,a) \mathbb{1}\left(d_h^{\mathsf{off}}(s,a) \geq \frac{2}{3} c_{\mathsf{off}}\left\{\frac{H^4 S^4 A^4 \log \frac{HSA}{\delta}}{N} + \frac{SA}{K}\right\}\right), \tag{82}$$

where the second line relies on (94). This combined with Lemma 1 results in

$$\sum_{(s,a) \in \mathcal{S} \times \mathcal{A}} \pi_h^t(a | s) \frac{\widehat{d}_h^{\mathsf{off}}(s,a)}{\frac{1}{KH} + \mathbb{E}_{\pi \sim \mu^{\mathsf{off}}}\left[\widehat{d}_h^{\pi}(s,a)\right]}$$

$$\leq \sum_{(s,a) \in \mathcal{S} \times \mathcal{A}} \pi_h^t(a | s) \frac{3 d_h^{\mathsf{off}}(s,a) \mathbb{1}\left(d_h^{\mathsf{off}}(s,a) \geq \frac{2}{3} c_{\mathsf{off}}\left\{\frac{H^4 S^4 A^4 \log \frac{HSA}{\delta}}{N} + \frac{SA}{K}\right\}\right)}{\frac{1}{KH} + \frac{1}{2}\mathbb{E}_{\pi \sim \mu^{\mathsf{off}}}\left[d_h^{\pi}(s,a) - 2e_h^{\pi}(s,a) - \frac{\xi}{4N}\right]}. \tag{83}$$

Moreover, inequality (56) tells us that: when $d_h^{\mathsf{off}}(s,a) \geq \frac{2}{3} c_{\mathsf{off}}\left(\frac{H^4 S^4 A^4 \log \frac{HSA}{\delta}}{N} + \frac{SA}{K^{\mathsf{on}}}\right)$ for some large enough constant $c_{\mathsf{off}} > 0$, we have

$$\mathbb{E}_{\pi \sim \mu^{\mathsf{off}}}\left[d_h^{\pi}(s,a) - 2e_h^{\pi}(s,a) - \frac{\xi}{4N}\right] \geq \mathbb{E}_{\pi \sim \mu^{\mathsf{off}}}\left[d_h^{\pi}(s,a) - \frac{4SA}{K^{\mathsf{on}}} - \frac{27 c_{\xi} S^4 A^4 H^4 \log \frac{HSA}{\delta}}{N}\right] \geq \frac{1}{2} d_h^{\mathsf{off}}(s,a). \tag{84}$$

In turn, this implies that

$$\sum_{(s,a) \in \mathcal{S} \times \mathcal{A}} \pi_h^t(a | s) \frac{3 d_h^{\mathsf{off}}(s,a) \mathbb{1}\left(d_h^{\mathsf{off}}(s,a) \geq \frac{2}{3} c_{\mathsf{off}}\left\{\frac{H^4 S^4 A^4 \log \frac{HSA}{\delta}}{N} + \frac{SA}{K}\right\}\right)}{\frac{1}{KH} + \frac{1}{2}\mathbb{E}_{\pi \sim \mu^{\mathsf{off}}}\left[d_h^{\pi}(s,a) - 2e_h^{\pi}(s,a) - \frac{\xi}{4N}\right]}$$

$$\leq \sum_{(s,a) \in \mathcal{S} \times \mathcal{A}} \pi_h^t(a | s) \frac{3 d_h^{\mathsf{off}}(s,a) \mathbb{1}\left(d_h^{\mathsf{off}}(s,a) \geq \frac{2}{3} c_{\mathsf{off}}\left\{\frac{H^4 S^4 A^4 \log \frac{HSA}{\delta}}{N} + \frac{SA}{K}\right\}\right)}{\frac{1}{KH} + \frac{1}{4} d_h^{\mathsf{off}}(s,a)}$$

$$\leq \sum_{(s,a) \in \mathcal{S} \times \mathcal{A}} \pi_h^t(a | s) \frac{3 d_h^{\mathsf{off}}(s,a)}{\frac{1}{4} d_h^{\mathsf{off}}(s,a)} = 12 S. \tag{85}$$

Consequently, combine (83) and (85) to yield

$$\sum_{h=1}^{H} \sum_{s \in \mathcal{S}} \mathbb{E}_{a \sim \pi_h^{t+1}(\cdot | s)} \left[ \frac{\widehat{d}_h^{\mathsf{off}}(s,a)}{\frac{1}{KH} + \mathbb{E}_{\pi \sim \mu^{\mathsf{off}}}\left[\widehat{d}_h^{\pi}(s,a)\right]} \right] \leq 12 SH, \tag{86}$$

which clearly validates the claim (81) (with an even better pre-constant).

Before moving forward, we single out one useful property that arises from the above arguments:

$$\mathbb{E}_{\pi \sim \mu^{\mathsf{off}}}\left[\widehat{d}_h^{\pi}(s,a)\right] \geq \frac{1}{12} \widehat{d}_h^{\mathsf{off}}(s,a). \tag{87}$$

To prove the validity of this claim (87), it suffices to make the following two observations:

- When $d_h^{\mathsf{off}}(s,a) \geq \frac{2}{3}c_{\mathsf{off}}\big(\frac{H^4 S^4 A^4 \log \frac{HSA}{\delta}}{N} + \frac{SA}{K^{\mathsf{on}}}\big)$, it has been shown in (83) and (84) in conjunction with (82) that

$$\mathbb{E}_{\pi \sim \mu^{\mathsf{off}}}\big[\widehat{d}_h^\pi(s,a)\big] \geq \frac{1}{4}d_h^{\mathsf{off}}(s,a) \geq \frac{1}{12}\widehat{d}_h^{\mathsf{off}}(s,a). \tag{88}$$

- When $d_h^{\mathsf{off}}(s,a) < \frac{2}{3}c_{\mathsf{off}}\big(\frac{H^4 S^4 A^4 \log \frac{HSA}{\delta}}{N} + \frac{SA}{K^{\mathsf{on}}}\big)$, one sees from (82) that $\widehat{d}_h^{\mathsf{off}}(s,a) = 0$, and hence (87) holds true trivially.

**Iteration complexity.** Suppose that the stopping criterion (37) is not yet met in the $k$-th iteration, namely,

$$\sum_{h=1}^{H}\sum_{s \in \mathcal{S}} \mathbb{E}_{a \sim \pi_h^{t+1}(\cdot|s)}\left[\frac{\widehat{d}_h^{\mathsf{off}}(s,a)}{\frac{1}{KH} + \mathbb{E}_{\pi' \sim \mu^{(k)}}\big[\widehat{d}_h^{\pi'}(s,a)\big]}\right] > 108SH. \tag{89}$$

It can be easily seen that

$$\sum_{h=1}^{H}\sum_{s \in \mathcal{S}} \mathbb{E}_{a \sim \pi_h^{t+1}(\cdot|s)}\left[\frac{\big(\frac{1}{KH} + \widehat{d}_h^{\pi^{(k)}}(s,a)\big)\widehat{d}_h^{\mathsf{off}}(s,a)}{\big(\frac{1}{KH} + \mathbb{E}_{\pi' \sim \mu^{(k)}}\big[\widehat{d}_h^{\pi'}(s,a)\big]\big)^2}\right]$$

$$\geq \sum_{h=1}^{H}\sum_{s \in \mathcal{S}} \mathbb{E}_{a \sim \pi_h^{t+1}(\cdot|s)}\left[\frac{\big(\frac{1}{KH} + \mathbb{E}_{\pi \sim \mu^{\mathsf{off}}}\big[\widehat{d}_h^\pi(s,a)\big]\big)\widehat{d}_h^{\mathsf{off}}(s,a)}{\big(\frac{1}{KH} + \mathbb{E}_{\pi' \sim \mu^{(k)}}\big[\widehat{d}_h^{\pi'}(s,a)\big]\big)^2}\right]$$

$$\geq \frac{1}{12}\sum_{h=1}^{H}\sum_{s \in \mathcal{S}} \mathbb{E}_{a \sim \pi_h^{t+1}(\cdot|s)}\left[\frac{\big(\widehat{d}_h^{\mathsf{off}}(s,a)\big)^2}{\big(\frac{1}{KH} + \mathbb{E}_{\pi' \sim \mu^{(k)}}\big[\widehat{d}_h^{\pi'}(s,a)\big]\big)^2}\right]$$

$$\geq 3\sum_{h=1}^{H}\sum_{s \in \mathcal{S}} \mathbb{E}_{a \sim \pi_h^{t+1}(\cdot|s)}\left[\frac{\widehat{d}_h^{\mathsf{off}}(s,a)}{\frac{1}{KH} + \mathbb{E}_{\pi' \sim \mu^{(k)}}\big[\widehat{d}_h^{\pi'}(s,a)\big]}\right]\mathbb{1}\left(\frac{\widehat{d}_h^{\mathsf{off}}(s,a)}{\frac{1}{KH} + \mathbb{E}_{\pi' \sim \mu^{(k)}}\big[\widehat{d}_h^{\pi'}(s,a)\big]} > 36\right)$$

$$= 3\sum_{h=1}^{H}\sum_{s \in \mathcal{S}} \mathbb{E}_{a \sim \pi_h^{t+1}(\cdot|s)}\left[\frac{\widehat{d}_h^{\mathsf{off}}(s,a)}{\frac{1}{KH} + \mathbb{E}_{\pi' \sim \mu^{(k)}}\big[\widehat{d}_h^{\pi'}(s,a)\big]}\right]$$

$$- 3\sum_{h=1}^{H}\sum_{s \in \mathcal{S}} \mathbb{E}_{a \sim \pi_h^{t+1}(\cdot|s)}\left[\frac{\widehat{d}_h^{\mathsf{off}}(s,a)}{\frac{1}{KH} + \mathbb{E}_{\pi' \sim \mu^{(k)}}\big[\widehat{d}_h^{\pi'}(s,a)\big]}\right]\mathbb{1}\left(\frac{\widehat{d}_h^{\mathsf{off}}(s,a)}{\frac{1}{KH} + \mathbb{E}_{\pi' \sim \mu^{(k)}}\big[\widehat{d}_h^{\pi'}(s,a)\big]} \leq 36\right)$$

$$\geq 3\sum_{h=1}^{H}\sum_{s \in \mathcal{S}} \mathbb{E}_{a \sim \pi_h^{t+1}(\cdot|s)}\left[\frac{\widehat{d}_h^{\mathsf{off}}(s,a)}{\frac{1}{KH} + \mathbb{E}_{\pi' \sim \mu^{(k)}}\big[\widehat{d}_h^{\pi'}(s,a)\big]}\right] - 108SH$$

$$\geq 2\sum_{h=1}^{H}\sum_{s \in \mathcal{S}} \mathbb{E}_{a \sim \pi_h^{t+1}(\cdot|s)}\left[\frac{\widehat{d}_h^{\mathsf{off}}(s,a)}{\frac{1}{KH} + \mathbb{E}_{\pi' \sim \mu^{(k)}}\big[\widehat{d}_h^{\pi'}(s,a)\big]}\right], \tag{90}$$

where the first inequality follows from the choice (32) of $\pi^{(k)}$, the second inequality is a consequence of the relation (87), and the last line makes use of (89). This in turn allows one to demonstrate that

$$\sum_{h=1}^{H}\sum_{s \in \mathcal{S}} \mathbb{E}_{a \sim \pi_h^{t+1}(\cdot|s)}\left[\frac{\big(\widehat{d}_h^{\pi^{(k)}}(s,a) - \mathbb{E}_{\pi' \sim \mu^{(k)}}\big[\widehat{d}_h^{\pi'}(s,a)\big]\big)\widehat{d}_h^{\mathsf{off}}(s,a)}{\big(\frac{1}{KH} + \mathbb{E}_{\pi' \sim \mu^{(k)}}\big[\widehat{d}_h^{\pi'}(s,a)\big]\big)^2}\right]$$

$$= \sum_{h=1}^{H}\sum_{s \in \mathcal{S}} \mathbb{E}_{a \sim \pi_h^{t+1}(\cdot|s)}\left[\frac{\big(\frac{1}{KH} + \widehat{d}_h^{\pi^{(k)}}(s,a)\big)\widehat{d}_h^{\mathsf{off}}(s,a)}{\big(\frac{1}{KH} + \mathbb{E}_{\pi' \sim \mu^{(k)}}\big[\widehat{d}_h^{\pi'}(s,a)\big]\big)^2}\right] - \sum_{h=1}^{H}\sum_{s \in \mathcal{S}} \mathbb{E}_{a \sim \pi_h^{t+1}(\cdot|s)}\left[\frac{\big(\frac{1}{KH} + \mathbb{E}_{\pi' \sim \mu^{(k)}}\big[\widehat{d}_h^{\pi'}(s,a)\big]\big)\widehat{d}_h^{\mathsf{off}}(s,a)}{\big(\frac{1}{KH} + \mathbb{E}_{\pi' \sim \mu^{(k)}}\big[\widehat{d}_h^{\pi'}(s,a)\big]\big)^2}\right]$$

$$\geq \sum_{h=1}^{H}\sum_{s \in \mathcal{S}} \mathbb{E}_{a \sim \pi_h^{t+1}(\cdot|s)}\left[\frac{\widehat{d}_h^{\mathsf{off}}(s,a)}{\frac{1}{KH} + \mathbb{E}_{\pi' \sim \mu^{(k)}}\big[\widehat{d}_h^{\pi'}(s,a)\big]}\right] > 108SH,$$

where the last line results from (90) and the condition (89). We can then readily derive

$$\sum_{h=1}^{H}\sum_{s \in \mathcal{S}} \mathbb{E}_{a \sim \pi_h^{t+1}(\cdot|s)}\left[\frac{\widehat{d}_h^{\mathsf{off}}(s,a)}{\frac{1}{KH} + \mathbb{E}_{\pi' \sim \mu^{(k)}}\big[\widehat{d}_h^{\pi'}(s,a)\big]}\right] - \sum_{h=1}^{H}\sum_{s \in \mathcal{S}} \mathbb{E}_{a \sim \pi_h^{t+1}(\cdot|s)}\left[\frac{\widehat{d}_h^{\mathsf{off}}(s,a)}{\frac{1}{KH} + \mathbb{E}_{\pi' \sim \mu^{(k+1)}}\big[\widehat{d}_h^{\pi'}(s,a)\big]}\right]$$

$$
\begin{aligned}
&= \sum_{h=1}^{H} \sum_{s \in \mathcal{S}} \mathbb{E}_{a \sim \pi_h^{t+1}(\cdot|s)} \left[ \frac{\left( \mathbb{E}_{\pi' \sim \mu^{(k+1)}} \left[ \widehat{d}_h^{\pi'}(s,a) \right] - \mathbb{E}_{\pi' \sim \mu^{(k)}} \left[ \widehat{d}_h^{\pi'}(s,a) \right] \right) \widehat{d}_h^{\mathsf{off}}(s,a)}{\left( \frac{1}{KH} + \mathbb{E}_{\pi' \sim \mu^{(k)}} \left[ \widehat{d}_h^{\pi'}(s,a) \right] \right) \left( \frac{1}{KH} + \mathbb{E}_{\pi' \sim \mu^{(k+1)}} \left[ \widehat{d}_h^{\pi'}(s,a) \right] \right)} \right] \\
&= \sum_{h=1}^{H} \sum_{s \in \mathcal{S}} \mathbb{E}_{a \sim \pi_h^{t+1}(\cdot|s)} \left[ \frac{\alpha \left( \widehat{d}_h^{\pi^{(k)}}(s,a) - \mathbb{E}_{\pi' \sim \mu^{(k)}} \left[ \widehat{d}_h^{\pi'}(s,a) \right] \right) \widehat{d}_h^{\mathsf{off}}(s,a)}{\left( \frac{1}{KH} + \mathbb{E}_{\pi' \sim \mu^{(k)}} \left[ \widehat{d}_h^{\pi'}(s,a) \right] \right) \left( \frac{1}{KH} + \mathbb{E}_{\pi' \sim \mu^{(k+1)}} \left[ \widehat{d}_h^{\pi'}(s,a) \right] \right)} \right] \\
&= \sum_{h=1}^{H} \sum_{s \in \mathcal{S}} \mathbb{E}_{a \sim \pi_h^{t+1}(\cdot|s)} \left[ \frac{\alpha \left( \widehat{d}_h^{\pi^{(k)}}(s,a) - \mathbb{E}_{\pi' \sim \mu^{(k)}} \left[ \widehat{d}_h^{\pi'}(s,a) \right] \right) \widehat{d}_h^{\mathsf{off}}(s,a)}{\left( \frac{1}{KH} + \mathbb{E}_{\pi' \sim \mu^{(k)}} \left[ \widehat{d}_h^{\pi'}(s,a) \right] \right)^2} \right] \\
&\quad - \sum_{h=1}^{H} \sum_{s \in \mathcal{S}} \mathbb{E}_{a \sim \pi_h^{t+1}(\cdot|s)} \left[ \frac{\alpha^2 \left( \widehat{d}_h^{\pi^{(k)}}(s,a) - \mathbb{E}_{\pi' \sim \mu^{(k)}} \left[ \widehat{d}_h^{\pi'}(s,a) \right] \right)^2 \widehat{d}_h^{\mathsf{off}}(s,a)}{\left( \frac{1}{KH} + \mathbb{E}_{\pi' \sim \mu^{(k)}} \left[ \widehat{d}_h^{\pi'}(s,a) \right] \right)^2 \left( \frac{1}{KH} + \mathbb{E}_{\pi' \sim \mu^{(k+1)}} \left[ \widehat{d}_h^{\pi'}(s,a) \right] \right)} \right] \\
&\geq 108 \alpha S H - \alpha^2 (K^3 H^4) = \frac{108 S^2}{K^3 H^2} - \frac{S^2}{K^3 H^2} = \frac{107 S^2}{K^3 H^2},
\end{aligned}
\tag{91}
$$

where the third line relies on the update rule (35), and the last line utilizes the choice (36) of $\alpha$.

In summary, the above argument reveals that: before the stopping criterion is met, each iteration is able to make progress at least as large as in (91). Recognizing the crude bound

$$
0 \leq \sum_{h=1}^{H} \sum_{s \in \mathcal{S}} \mathbb{E}_{a \sim \pi_h^{t+1}(\cdot|s)} \left[ \frac{\widehat{d}_h^{\mathsf{off}}(s,a)}{\frac{1}{KH} + \mathbb{E}_{\pi' \sim \mu} \left[ \widehat{d}_h^{\pi'}(s,a) \right]} \right] \leq KH \sum_{h=1}^{H} \sum_{s \in \mathcal{S}} \mathbb{E}_{a \sim \pi_h^{t+1}(\cdot|s)} \left[ \widehat{d}_h^{\mathsf{off}}(s,a) \right] \leq KH^2
$$

that holds for any $\mu \in \Delta(\Pi)$, one can combine this with (91) to conclude that: the proposed procedure terminates within $O\left( \frac{K^4 H^4}{S^2} \right)$ iterations, as claimed.

## G  Proofs of technical lemmas

### G.1  Proof of Lemma 2

The Bernstein inequality combined with the union bound tells us that, with probability at least $1 - \delta/3$,

$$
\begin{aligned}
\left| \frac{2 N_h^{\mathsf{off}}(s,a)}{K^{\mathsf{off}}} - d_h^{\mathsf{off}}(s,a) \right| &\leq 6 \sqrt{\frac{d_h^{\mathsf{off}}(s,a) \log \frac{HSA}{\delta}}{K^{\mathsf{off}}}} + \frac{6 \log \frac{HSA}{\delta}}{K^{\mathsf{off}}} \\
&\leq \frac{d_h^{\mathsf{off}}(s,a)}{2} + \frac{24 \log \frac{HSA}{\delta}}{K^{\mathsf{off}}}
\end{aligned}
\tag{92}
$$

holds simultaneously for all $(s,a,h) \in \mathcal{S} \times \mathcal{A} \times [H]$, where the last line invokes the AM-GM inequality. This in turn reveals that

$$
\frac{4 N_h^{\mathsf{off}}(s,a)}{3 K^{\mathsf{off}}} - \frac{16 \log \frac{HSA}{\delta}}{K^{\mathsf{off}}} \leq d_h^{\mathsf{off}}(s,a) \leq \frac{4 N_h^{\mathsf{off}}(s,a)}{K^{\mathsf{off}}} + \frac{48 \log \frac{HSA}{\delta}}{K^{\mathsf{off}}}.
\tag{93}
$$

As a result, we can show that:

- If $\frac{N_h^{\mathsf{off}}(s,a)}{K^{\mathsf{off}}} \geq c_{\mathsf{off}} \left( \frac{\log \frac{HSA}{\delta}}{K^{\mathsf{off}}} + \frac{H^4 S^4 A^4 \log \frac{HSA}{\delta}}{N} + \frac{SA}{K^{\mathsf{on}}} \right)$ for some $c_{\mathsf{off}} \geq 48$, then one has

$$
\frac{2 N_h^{\mathsf{off}}(s,a)}{3 K^{\mathsf{off}}} \leq d_h^{\mathsf{off}}(s,a) \leq \frac{6 N_h^{\mathsf{off}}(s,a)}{K^{\mathsf{off}}} \qquad \text{and} \qquad \widehat{d}_h^{\mathsf{off}}(s,a) = \frac{2 N_h^{\mathsf{off}}(s,a)}{K^{\mathsf{off}}}
\tag{94a}
$$

$$
\implies \qquad \frac{1}{3} \widehat{d}_h^{\mathsf{off}}(s,a) \leq d_h^{\mathsf{off}}(s,a) \leq 3 \widehat{d}_h^{\mathsf{off}}(s,a).
\tag{94b}
$$

- If instead $\frac{N_h^{\mathsf{off}}(s,a)}{K^{\mathsf{off}}} < c_{\mathsf{off}} \left( \frac{\log \frac{HSA}{\delta}}{K^{\mathsf{off}}} + \frac{H^4 S^4 A^4 \log \frac{HSA}{\delta}}{N} + \frac{SA}{K^{\mathsf{on}}} \right)$, then one has $\widehat{d}_h^{\mathsf{off}}(s,a) = 0$, and therefore,

$$
d_h^{\mathsf{off}}(s,a) \geq 0 = \frac{1}{3} \widehat{d}_h^{\mathsf{off}}(s,a),
$$

$$d_h^{\text{off}}(s,a) \le \frac{4N_h^{\text{off}}(s,a)}{K^{\text{off}}} + \frac{48\log\frac{HSA}{\delta}}{K^{\text{off}}} < 5c_{\text{off}}\left\{\frac{\log\frac{HSA}{\delta}}{K^{\text{off}}} + \frac{H^4S^4A^4\log\frac{HSA}{\delta}}{N} + \frac{SA}{K^{\text{on}}}\right\}$$

$$= \widehat{d}_h^{\text{off}}(s,a) + 5c_{\text{off}}\left\{\frac{\log\frac{HSA}{\delta}}{K^{\text{off}}} + \frac{H^4S^4A^4\log\frac{HSA}{\delta}}{N} + \frac{SA}{K^{\text{on}}}\right\}.$$

Taken collectively, these inequalities demonstrate that

$$\frac{1}{3}\widehat{d}_h^{\text{off}}(s,a) \le d_h^{\text{off}}(s,a) \le \widehat{d}_h^{\text{off}}(s,a) + 5c_{\text{off}}\left\{\frac{\log\frac{HSA}{\delta}}{K^{\text{off}}} + \frac{H^4S^4A^4\log\frac{HSA}{\delta}}{N} + \frac{SA}{K^{\text{on}}}\right\}, \quad (95)$$

provided that $c_{\text{off}} > 0$ is sufficiently large.

### G.2    Proof of Lemma 6

Before embarking on the proof, let us introduce several notation. For each $1 \le h \le H$, define $P_h^{\pi^\star} \in \mathbb{R}^{S\times S}$ and $d_h^{\pi^\star} \in \mathbb{R}^S$ such that: for all $s \in \mathcal{S}$,

$$P_h^{\pi^\star}(s,\cdot) = P_h\big(\cdot\,|s,\pi^\star(s)\big) \qquad \text{and} \qquad d_h^{\pi^\star}(s) = d_h^{\pi^\star}\big(s,\pi^\star(s)\big). \quad (96)$$

To begin with, it can be easily seen from (69) and the basic fact $\mathsf{Var}_{P_h(\cdot|s,a)}\big(\widehat{V}_{h+1}\big) \le H^2$ that

$$\big\langle d_j^{\pi^\star}, V_j^\star - V_j^{\widehat\pi}\big\rangle \le \underbrace{8H\sum_{h:h\ge j}\sum_s \max_{a:(s,a)\in\mathcal{I}_h}\sqrt{2d_h^\star(s,a)C^\star(\sigma)\widehat{d}_h^{\text{off}}(s,a)}\sqrt{\frac{c_{\mathsf{b}}\log\frac{K}{\delta}}{N_h^{\text{trim}}(s,a)+1}}}_{=:\gamma_3}$$

$$+ \underbrace{4H\sum_{h:h\ge j}\sum_{(s,a)\in\mathcal{T}_h}\widehat{d}_h^{\pi^\star}(s,a)\sqrt{\frac{c_{\mathsf{b}}\log\frac{K}{\delta}}{N_h^{\text{trim}}(s,a)+1}}}_{=:\gamma_2} + \frac{61c_\xi H^6 S^4 A^4}{N}\log\frac{K}{\delta} \quad (97)$$

holds for any $j \in [H]$. Note that we have bounded $\gamma_2$ in (79). We then need to bound $\gamma_3$.

With regards to the term $\gamma_3$: invoking similar arguments as for (72) leads to

$$\gamma_3 \le 64c_{\mathsf{b}}H\sum_{h:h\ge j}\sum_s \max_a\sqrt{2d_h^{\pi^\star}(s,a)C^\star(\sigma)\widehat{d}_h^{\text{off}}(s,a)}\sqrt{\frac{\log\frac{K}{\delta}}{1/H + \frac{1}{72}K^{\text{on}}\mathbb{E}_{\pi\in\mu^{\text{imitate}}}\big[\widehat{d}_h^\pi(s,a)\big]}} + 4H^2SC^\star(\sigma)\Big(\frac{\xi}{N} + \frac{1}{K^{\text{on}}H}\Big)$$

$$\le 64c_{\mathsf{b}}H\sqrt{2\sum_{h:h\ge j}\sum_{s,a}d_h^{\pi^\star}(s,a)}\cdot\sqrt{\sum_{h:h\ge j}\sum_s\max_a\frac{C^\star(\sigma)\widehat{d}_h^{\text{off}}(s,a)\log\frac{K}{\delta}}{1/H + \frac{1}{72}K^{\text{on}}\mathbb{E}_{\pi\in\mu^{\text{imitate}}}\big[\widehat{d}_h^\pi(s,a)\big]}} + 4H^2SC^\star(\sigma)\Big(\frac{\xi}{N} + \frac{1}{K^{\text{on}}H}\Big)$$

$$\le 768c_{\mathsf{b}}\sqrt{\frac{13H^4SC^\star(\sigma)}{K^{\text{on}}}\log\frac{K}{\delta}} + 4H^2SC^\star(\sigma)\Big(\frac{\xi}{N} + \frac{1}{K^{\text{on}}H}\Big), \quad (98)$$

where the second step invokes the Cauchy-Schwartz inequality, and the last line comes from (67). Similarly, repeating the above argument but focusing on the offline dataset, we can derive (which we omit for the sake of brevity)

$$\gamma_3 \le 64c_{\mathsf{b}}H\sum_{h:h\ge j}\sum_s \max_a\sqrt{2d_h^{\pi^\star}(s,a)C^\star(\sigma)\widehat{d}_h^{\text{off}}(s,a)}\sqrt{\frac{\log\frac{K}{\delta}}{1/H + \frac{1}{72}K^{\text{off}}d_h^{\text{off}}(s,a)]}} + 4H^2SC^\star(\sigma)\Big(\frac{\xi}{N} + \frac{1}{K^{\text{off}}H}\Big)$$

$$\le 64c_{\mathsf{b}}H\sqrt{2\sum_{h:h\ge j}\sum_{s,a}d_h^{\pi^\star}(s,a)}\cdot\sqrt{\sum_{h:h\ge j}\sum_s\max_a\frac{C^\star(\sigma)\widehat{d}_h^{\text{off}}(s,a)\log\frac{K}{\delta}}{1/H + \frac{1}{72}K^{\text{off}}d_h^{\text{off}}(s,a)}} + 4H^2SC^\star(\sigma)\Big(\frac{\xi}{N} + \frac{1}{K^{\text{off}}H}\Big)$$

$$\le 768c_{\mathsf{b}}\sqrt{\frac{6H^4SC^\star(\sigma)}{K^{\text{off}}}\log\frac{K}{\delta}} + 4H^2SC^\star(\sigma)\Big(\frac{\xi}{N} + \frac{1}{K^{\text{off}}H}\Big), \quad (99)$$

where the last line makes use of (86).

Combining (98), (99) and (79) with (97), we can show that

$$\langle d_j^{\pi^\star}, V_j^\star - V_j^{\widehat{\pi}} \rangle \le \min \left\{ 768c_{\mathsf{b}} \sqrt{\frac{13H^4 SC^\star(\sigma)}{K^{\mathsf{on}}} \log \frac{K}{\delta}} + 4H^2 SC^\star(\sigma)\Big(\frac{\xi}{N} + \frac{1}{KH}\Big), \ 768c_{\mathsf{b}} \sqrt{\frac{6H^4 SC^\star(\sigma)}{K^{\mathsf{off}}} \log \frac{K}{\delta}} \right\}$$

$$+ 96c_{\mathsf{b}} \sqrt{\frac{2H^3 SA(2\widehat{\sigma} + HS\xi/N)}{K^{\mathsf{on}}} \log \frac{HK}{\delta}} + 436H^2 SA\Big(\frac{\xi}{N} + \frac{1}{\min\{K^{\mathsf{on}}, K^{\mathsf{off}}\}H}\Big)$$

$$+ \frac{61c_\xi H^6 S^4 A^4}{N} \log \frac{K}{\delta}$$

$$\le 1 \tag{100}$$

for all $1 \le j \le H$, with the proviso that

$$\frac{H^7 S^5 A^4}{K^{\mathsf{on}}} \log^2 \frac{K}{\delta} \le c_{10}$$

$$\frac{H^5 S^4 A^3 C^\star(\sigma) \log \frac{K}{\delta}}{K^{\mathsf{on}}} \le c_{10}$$

$$\frac{HSC^\star(\sigma) \log \frac{K}{\delta}}{K^{\mathsf{off}}} \le c_{10}$$

$$\frac{HSA \log \frac{K}{\delta}}{K^{\mathsf{off}}} \le c_{10}$$

for some sufficiently small constant $c_{10} > 0$. As a consequence, we can demonstrate that

$$\sum_{h:h \ge j} \sum_{s,a} d_h^{\pi^\star}(s,a) \mathsf{Var}_{P_h(\cdot|s,a)}(\widehat{V}_{h+1})$$

$$\le \sum_{h:h \ge j} 2 \sum_{s,a} d_h^{\pi^\star}(s,a) \Big( \mathsf{Var}_{P_h(\cdot|s,a)}(V_{h+1}^\star) + \mathsf{Var}_{P_h(\cdot|s,a)}(V_{h+1}^\star - \widehat{V}_{h+1}) \Big)$$

$$\le 4H^2 + H \sum_{h:h \ge j} \sum_s d_h^{\pi^\star}(s, \pi^\star(s)) \mathbb{E}_{P_h(\cdot|s,\pi^\star(s))} \big[ V_{h+1}^\star - \widehat{V}_{h+1} \big]$$

$$= 4H^2 + H \sum_{h:h \ge j} (d_h^{\pi^\star})^\top P_h^{\pi^\star} \big[ V_{h+1}^\star - \widehat{V}_{h+1} \big]$$

$$= 4H^2 + H \sum_{h \ge j} (d_{h+1}^{\pi^\star})^\top \big[ V_{h+1}^\star - \widehat{V}_{h+1} \big] \le 5H^2 \tag{101}$$

for all $1 \le j \le H$. Here, the third line in (101) applies the following fact

$$\sum_h \sum_{s,a} d_h^{\pi^\star}(s,a) \mathsf{Var}_{P_h(\cdot|s,a)}(V_{h+1}^\star)$$

$$= \sum_h \sum_s d_h^{\pi^\star}(s, \pi^\star(s)) \Big[ \big\langle P_h^{\pi^\star}(s,\cdot), V_{h+1}^\star \circ V_{h+1}^\star \big\rangle - \big( \big\langle P_h^{\pi^\star}(s,\cdot), V_{h+1}^\star \big\rangle \big)^2 \Big]$$

$$= \sum_h \sum_s d_h^{\pi^\star}(s, \pi^\star(s)) \Big[ \big\langle P_h^{\pi^\star}(s,\cdot), V_{h+1}^\star \circ V_{h+1}^\star \big\rangle - \big( V_h^\star(s) - r_h^\star(s, \pi^\star(s)) \big)^2 \Big]$$

$$\le \sum_h \sum_s d_h^{\pi^\star}(s, \pi^\star(s)) \Big[ \big\langle P_h^{\pi^\star}(s,\cdot), V_{h+1}^\star \circ V_{h+1}^\star \big\rangle - \big( V_h^\star(s) \big)^2 + 2H \Big]$$

$$= \sum_h (d_h^{\pi^\star})^\top P_h^{\pi^\star} \big( V_{h+1}^\star \circ V_{h+1}^\star \big) - \sum_h (d_h^{\pi^\star})^\top \big( V_h^\star \circ V_h^\star \big) \Big] + 2H^2$$

$$= \sum_h (d_{h+1}^{\pi^\star})^\top \big( V_{h+1}^\star \circ V_{h+1}^\star \big) - \sum_h (d_h^{\pi^\star})^\top \big( V_h^\star \circ V_h^\star \big) + 2H^2$$

$$\le 2H^2,$$

where the second identity comes from the Bellman equation; the third relation uses the fact that $V_h^\star(s) \le H$, and the penultimate line holds since $(d_h^{\pi^\star})^\top P_h^{\pi^\star} = (d_{h+1}^{\pi^\star})^\top$; the penultimate step in (101) is due to (100); and the line in (101) holds true since $(d_h^{\pi^\star})^\top P_h^{\pi^\star} = (d_{h+1}^{\pi^\star})^\top$.

