# OpenReview forum: "Reward-agnostic Fine-tuning: Provable Statistical Benefits of Hybrid Reinforcement Learning"
_NeurIPS.cc/2023/Conference — NeurIPS 2023 poster_

### Official Review · Reviewer_sEvV · 2023-07-06

**Soundness:** 3 good
**Presentation:** 3 good
**Contribution:** 3 good
**Rating:** 6
**Confidence:** 2

**Summary:**

The authors prove that augmenting an offline dataset with online data collection leads to improved sample complexity compared to online and offline RL in the tabular setting, under the condition of single-policy partial concentrability. This is an extension of previous work on single-policy concentrability, relaxing the requirement that the offline dataset cover the full state-action space of the optimal policy. The authors propose a three step algorithm, given a budget of $K^{on}$ online data collection steps - spend the first third estimating occupancy distributions, the next two thirds on imitating the provided dataset and an exploration focused policy respectively, and finally finding a policy using standard offline RL on the original and collected datasets. This algorithm can be run in polynomial time using a Frank-Wolfe approximation to find the optimal policies in the second step.

**Strengths:**

- The algorithm and proof are both novel to the best of my knowledge, and address an important question in RL theory.
- The result is a general improvement over previous work, relaxing the coverage requirements without imposing new constraints or significant additional computation.
- The paper is generally well written without many typos or errors.

**Weaknesses:**

- Some of the choices in the algorithm could be better justified in the main text. Without referring to the Appendix the motivation for the imitation and exploration datasets is difficult to understand.
- While the Frank-Wolfe updates proposed include a polynomial stopping rule, it's not clear how feasible this bound would be in any realistic task.

Minor:
- Section 2 (Line 123): noation -> notation

**Questions:**

- What is the definition of $f_{mixed}$ in section 4?
- Intuitively, what do the terms $\gamma_1$ and $\gamma_2$ represent?

**Limitations:**

Yes

---

> ### Author Rebuttal · Authors · 2023-08-10
>
> Thanks a lot for the helpful comments and valuable feedback.
> Please find below our point-to-point response, which will also be incorporated into the final paper.
>
> 1. In the original submission, the main algorithm and several associated discussions were included in the appendix due to the page limit.
> In the final version, we believe that we will have more space, and will definitely include the main algorithm and more discussion in the main text.
>
> 2. We would like to note that: while this polynomial stopping looks complicated, it is actually easily computable and checkable.
> In addition, we have also discussed the feasibility of the bound Eq. 36 for solving Eq. 20b in Appendix F.
> Note that our paper is mainly to show that there exists a polynomial time algorithm that can achieve provable statistical benefits; in practice, there could be more flexible stopping rules that achieve even more appealing performance.
>
>
> 3. We note that $f_{\mathsf{mixed}}$ is defined on the left-hand side of Eq. 26, which is given by
> \begin{align}
> f_{\mathsf{mixed}} := \frac{H^3SA\min\\{H\sigma, 1\\}}{\varepsilon^2} + \frac{H^3SC^{\star}(\sigma)}{\varepsilon^2}.
> \end{align}
> We will make this definition more clear in the final paper.
>
> 4. Informally, $\gamma_1$ represents the error from the state-action pairs that are well explored by offline dataset,
> and $\gamma_2$ represents the error from the state-action pairs that are poorly covered by offline dataset.
>
> Reference:
>
> [Li et al. 2023] Li, G., Yan, Y., Chen, Y., and Fan, J. (2023). Minimax-optimal reward-agnostic exploration in reinforcement learning. arXiv preprint arXiv:2304.0727.

---

> > ### Comment · Reviewer_sEvV · 2023-08-22
> >
> > Thank you to the authors for addressing my questions

---

### Official Review · Reviewer_Vg7V · 2023-07-06

**Soundness:** 3 good
**Presentation:** 4 excellent
**Contribution:** 3 good
**Rating:** 7
**Confidence:** 3

**Summary:**

The paper proposes a new three step algorithm in the hybrid RL setting for tabular MDPs that improves in terms of sample complexity on the pure online and as well as the pure offline setting. The notion of single-policy partial concentrability is introduced which offers an intuitive concept that connects the sample complexity of pure online and pure offline RL. The algorithm leverages results from reward agnostic exploration to explore the parts of the state-action space that are missing from the offline dataset and then applies an offline learning algorithm to  find an $\epsilon$-suboptimal policy. The sample complexity result is compared to existing results and the algorithms advantages and limitations are discussed.

**Strengths:**

The paper is well written and reads itself easily, the algorithm and the sample complexity result are introduced in a digestible manner without loss in rigor.

The concept of single-policy partial concentrability captures the two problems with offline datasets well (lack of coverage and mismatch with the state-action distribution of the optimal policy) and provides a neat connection between sample complexity results of pure online and pure offline approaches.

For the tabular setting without function approximation the sample complexity result is SOTA.

The advantages and limitations of the algorithms are extensively discussed.

**Weaknesses:**

The motivation/intuition why the number of samples coming from the offline dataset distribution needs to be increased with the imitation exploration policy is not clear to me. Why is the offline dataset and the dataset collected with $\pi^{explore}$ not enough ?

I find the notation used for the single-policy partial concentrability coefficient slightly confusing:

- In equation 10 I would write $\\{G_{h}\\}_{1 \leq h \leq H} \in G(\sigma)$
- in equation 11 I would write $G(\sigma) \coloneqq \\{ \\{ G_{h}\\}_{1 \leq h \leq H} | \forall 1 \leq h \leq H , G_h \subseteq S \times A, ... \\} $

**Questions:**

How does the sample complexity result from "Hybrid RL: Using Both Offline and Online Data Can Make RL Efficient" (Song et al., 2023) compare to the presented result? The mentioned paper looks at bilinear models so similarly to Wagenmaker and Pacchiano is not specifically designed for the tabular case.

**Limitations:**

The limitations are discussed. An empirical comparison of the presented algorithm with other hybrid RL algorithms would have been interesting but seems out of scope of the presented work.

Negative Societal Impact - Not Applicable

---

> ### Author Rebuttal · Authors · 2023-08-10
>
> Thanks a lot for the helpful comments and valuable feedback.
> Please find below our point-to-point response, which will also be incorporated into the final paper.
>
> 1. If the number of the newly explored samples is smaller than the number of offline data samples, then the offline dataset and the dataset collected using $\pi^{\mathsf{explore}}$ are enough.
> In comparison, if the number of the newly explored samples is larger than the number of offline data samples, then the offline dataset is not enough in achieving our performance bounds.
> For instance, if $C^{\star}(\sigma) < A$ for some $\sigma < 1/H$, then we need the newly explored dataset to include some imitation effort in order to enlarge the offline dataset and achieve better performance.
> We will make this point clear in the revised paper.
>
> 2. Thanks for pointing out the notational issue, which we will revise accordingly in the revised paper.
>
> 3. We have compared our result with this reference in Appendix A with the following text ``Song et al. (2022) mainly focused on the issue of computational efficiency, and the algorithm proposed therein does not come with improved sample complexity.''
> We will add more discussion and comparison with this reference in the revised paper.
>
> 4. Yes, our current work is mainly our theoretical attempt towards understanding the provable benefits of hybrid setting.
> While the current algorithm is not yet practically efficient,
> we hope that our design principle might shed light on the design of more practical yet sample-efficient solutions.

---

> > ### Comment · Reviewer_Vg7V · 2023-08-18
> >
> > Thanks for addressing all my comments/questions

---

### Official Review · Reviewer_R4EB · 2023-07-07

**Soundness:** 3 good
**Presentation:** 3 good
**Contribution:** 3 good
**Rating:** 7
**Confidence:** 3

**Summary:**

This paper studies hybrid reinforcement learning under tabular MDPs, where the agent has access to both a given offline dataset and online interaction with environment. This paper first proposes an interesting notion of single-policy partial concentrability, which relaxs previous concentrability assumption and accounts for the trade offs between distributional mismatch and partial converage. Then this paper proposes a novel three stage algorithm, which can automatically identify the optiaml trade offs. Then the paper provides a sample complexity upper bound. The sample complexity result shows the benefit of hybrid RL compared with pure online or offline RL.

**Strengths:**

1. The notion of single-policy partial concentrability is interesting and reasonable to me and relaxs previous concentrability assumption.

2. This paper proposes a novel three-stage algorithm for hybrid RL, which can automatically identify the optiaml trade offs between distributional mismatch and partial converage. Intuitively, the agent can imitate the offline dataset to collect partial informative data, and explore the unknown data which is not covered by offline dataset.

3. When compared to pure online or offline results, the samplexity complexity result shows the benefit of hybrid RL.



**Weaknesses:**

1. This paper mentions Frank-Wolfe-type subroutine in introduction and conclusion in main body. However the Frank-Wolfe-type subroutine is even not discussed in main body. Also, pseudocode of algorithm 1 is deferred in Appendix, which make it not easy to read and follow the idea of algorithm.

2. This paper only provides theoretical results without experiment results.

**Questions:**

As listed in weakness.

**Limitations:**

There is no negative impact in this work.

---

> ### Author Rebuttal · Authors · 2023-08-10
>
> Thanks a lot for the helpful comments and valuable feedback.
> Please find below our point-to-point response, which will also be incorporated into the final paper.
>
> 1. Frank-Wolfe-type method is an efficient first-order iterative algorithm to solve constrained convex optimization problems. We will make sure to introduce it in more detail in the final version. In the original submission, the main algorithm was included in the appendix due to the page limit.
> In the final version, we believe that we will have more space, and will definitely include the main algorithm in the main text.
>
>
> 2. This work is mainly our theoretical attempt towards understanding the provable benefits of hybrid setting.
> While the current algorithm is not yet practically efficient,
> we hope that our design principle might shed light on the design of more practical yet sample-efficient solutions.

---

> > ### Comment · Reviewer_R4EB · 2023-08-19
> >
> > Thank the authors for addressing my comments, I will maintain my score.

---

### Official Review · Reviewer_GVvQ · 2023-07-07

**Soundness:** 3 good
**Presentation:** 3 good
**Contribution:** 3 good
**Rating:** 6
**Confidence:** 2

**Summary:**

This paper investigates tabular reinforcement learning (RL) in a hybrid setting, where both an offline dataset and online interactions with an unknown environment are available. The main question addressed is how to effectively use online data to enhance and complement the offline dataset for policy fine-tuning. The authors propose a three-stage hybrid RL algorithm that outperforms pure offline RL and pure online RL in terms of sample complexity. The algorithm does not require reward information during data collection.


**Strengths:**

- Innovative approach: The paper introduces a novel three-stage hybrid RL algorithm that combines online and offline data to improve sample complexity in tabular RL.
- Theoretical contributions: The authors develop a new notion, single-policy partial concentrability, which captures the trade-off between distribution mismatch and lack of coverage. They provide theoretical results that demonstrate the statistical benefits of hybrid RL compared to pure online and offline RL.


**Weaknesses:**

- The paper provides no empirical evaluations or demonstrations of the proposed algorithm. Neither does it shed light on the design of practical algorithms.
- There remain some issues unsolved in the paper. See the questions for details.



**Questions:**

- Empirical estimation of the occupancy distribution in Sec. 3.1 can have high variance. How does the proposed approach handle this issue?
- Why do we need to imitate the offline dataset? What if the dataset is collected with random policies?
- The analyses are conducted in the tabular case? How to extend them to continuous state/action spaces?


**Limitations:**

Yes

---

> ### Author Rebuttal · Authors · 2023-08-10
>
> Thanks a lot for the helpful comments and valuable feedback.
> Please find below our point-to-point response, which will also be incorporated into the final paper.
>
> 1. This work is mainly our theoretical attempt towards understanding the provable benefits of hybrid setting.
> While the current algorithm is not yet practically efficient,
> we hope that our design principle might shed light on the design of more practical yet sample-efficient solutions.
>
> 2. We use the estimation method proposed in Li et al. 2023, which is completed through estimating the transition probability model $P_h$ by exploring all state-action pairs.
> This approach turns out to be sufficient in handling the variance issue.
>
> 3. In some applications, the offline dataset might be obtained by "experts'', meaning that it might have (much) higher quality than random dataset.
> If this happens, it becomes beneficial for us to imitate the behavior of offline dataset.
> As a more technical example, if $C^{\star}(\sigma)$ defined in Eq. 10 is small (corresponding to near-expert data), then we can benefit through imitating the offline dataset (see the discussion below our main theorem).
> On the other hand, we also agree that if the dataset is collected with random policies and when $C^{\star}(\sigma)$ is large, there is no need to imitate the offline dataset.
> It boils down to how to balance the effort spent on imitation and the effort spent on exploration.
> Our scheme turns out to be provably order-wide sample efficient for this purpose.
>
>
> 4. Yes, we focus on the tabular case here.
> One way to deal with the continuous state/action spaces is to resort to function approximation.
> It is possible to extend the idea developed in this work to accommodate linear MDPs.
> There are several important steps needed. Firstly, we need to find an efficient estimation method for the transition kernels of  the linear MDP. Secondly, we need to use the estimated MDP to imitate the behavior policy of the offline dataset and also learn a good exploration policy.
> Finally, we need to invoke (or develop) a sample-optimal offline RL algorithm in the case of linear MDPs to find the optimal policy based on the collected sample trajectories.
> Accomplishing all these details needs substantial efforts, which we leave for future work.
>
>
> Reference:
>
> [Li et al. 2023] Li, G., Yan, Y., Chen, Y., and Fan, J. (2023). Minimax-optimal reward-agnostic exploration in reinforcement learning. arXiv preprint arXiv:2304.0727.

---

> > ### Comment · Reviewer_GVvQ · 2023-08-19
> > **Response**
> >
> > I acknowledge the authors' rebuttal and I remain my rating towards acceptance.

---

### Official Review · Reviewer_Ukmm · 2023-07-19

**Soundness:** 2 fair
**Presentation:** 3 good
**Contribution:** 3 good
**Rating:** 6
**Confidence:** 3

**Summary:**

This paper studies a tabular reinforcement learning setting where the agent tries to learn the optimal policy by both offline data (obtained by exploiting the optimal policy) and online exploration. A three-stage algorithm is proposed to obtain such a policy, which has lower sample complexity compared to pure online and pure offline reinforcement learning algorithms.

**Strengths:**

This paper proposes the first algorithm for the hybrid RL setting that can achieve lower sample complexity than pure online and pure offline algorithms, which is innovative and inspiring. Overall, The writing is clear.

**Weaknesses:**

There are a few crucial technique details that are not clear enough, which should be well justified. Please see my questions below.

**Questions:**

Line 120: what do the authors mean by a "unknown" initial state distribution? Does its value depend on specific applications?

Line 126-127: What is the difference between a mixed deterministic policy and a randomized one? Could the author provide a small example to elucidate their difference?

Definition 1: I think the definition of $C^\star$ is associated with some policy $\pi^\star$. If so, the authors should specify this.

Line 195: How should one decide which samples belong to the first/second half? Or can we divide the two halves arbitrarily?

In the "Imitating the offline dataset" step, the authors appear to assume that the offline dataset is obtained by executing an optimal policy. (1) If so, I think the authors should formalize this assumption. (2) Also, this assumption is strong (where the information of the optimality is leaked to the decision maker). Could the authors justify this assumption?

It feels fishy that authors claimed that "Towards this end, it suffices to invoke the reward-agnostic online exploration scheme" in line 250, while they assumed that "the reward function is fully revealed upon completion of online data collection" in line 118.

**Limitations:**

Please see the weaknesses above.

---

> ### Author Rebuttal · Authors · 2023-08-10
>
> Thanks a lot for the helpful comments and valuable feedback.
> Please find below our point-to-point response, which will also be incorporated into the final paper.
>
> 1.  When we say the initial state distribution is ``unknown'', we mean that this distribution is not revealed to the learner {\em a priori}.
> And yes, the unknown initial state distribution could depend on specific applications. We will make these more clear in the final paper.
>
> 2. For the bandit case with horizon length $H = 1$, a mixed deterministic policy is equivalent to a randomized one.
> When $H > 1$, these two cases are not equivalent.
> For example, the mixed deterministic policy $\mu(\pi^1) = \mu(\pi^2) = \frac{1}{2}$ has no corresponding randomized one,
> where
> \begin{align}
> \pi_1^1(a_0 | s) = 1,
> \pi_2^1(a_0 | s) = 1,
> \qquad\text{and}\qquad
> \pi_1^1(a_1 | s) = 1,
> \pi_2^1(a_1 | s) = 1,
> \end{align}
> for $S = 1, A = 2, H = 2$ with $P(s | 0, \cdot) = 1$,
> which yields $P_{\mu}(\{s, a_0, s, a_0, s\}) = P_{\mu}(\{s, a_1, s, a_1, s\}) = \frac{1}{2}$.
> We will discuss these differences in the final paper.
>
> 3. Yes, $C^{\star}$ is defined with respect to some $\pi^{\star}$. We will specify this more clearly in the final paper.
>
> 4. As defined in Eq. 12 in the original paper, $\mathcal{D}^{\mathsf{off}, 1}$ (resp. $\mathcal{D}^{\mathsf{off}, 2}$) consists of the first (resp. last) $K^{\mathsf{off}}/2$ independent trajectories from $\mathcal{D}^{\mathsf{off}}$. And yes, we can divide the two halves arbitrarily and independently.
>
> 5. Thanks for raising this point. It is noteworthy that we didn't assume that the offline dataset is obtained by executing an optimal policy.
> In fact, we impose no requirement on the offline dataset, and our algorithm works for an arbitrarily given dataset.
> We shall make these points more clear in the revised paper.
>
> 6. Note that in prior RL literature, "reward-agnostic'' or "reward-free'' refers to the case where online data collection does not utilize any reward information (but reward information is revealed in the final policy learning stage). This is consistent with our setting and algorithm: in Stage 1 and 2, we do not utilize any information about the reward, while reward information is only provided in Stage 3 to compute the optimal policy via offline RL. We will make these terminology more clear in the final paper.

---

> > ### Comment · Reviewer_Ukmm · 2023-08-14
> >
> > Thank you for addressing my comments! I will raise my rating.

---

> > > ### Author Response · Authors · 2023-08-14
> > >
> > > Thanks for kindly raising the score. We will definitely revise the paper accordingly to address all your comments.

---

### Official Review · Reviewer_Lcd6 · 2023-07-21

**Soundness:** 3 good
**Presentation:** 2 fair
**Contribution:** 2 fair
**Rating:** 4
**Confidence:** 3

**Summary:**

This paper studies hybrid RL (online RL with offline data) in the specialized setting of tabular MDPs, while providing tighter bounds than prior works. Namely, they seem to propose a novel and weaker concentrability notion called single-policy partial concentrability. The algorithm takes three stages: 1. estimate occupancy distribution, 2. perform online exploration, 3. invoke offline RL for policy learning.

**Strengths:**

1. A weaker notion of partial coverage for hybrid RL is proposed in Def 2.
2. While I'm not familiar with all previous hybrid RL works, this paper seems to be one of the first that obtains better sample complexity in online RL, in the tabular setting.

**Weaknesses:**

1. While the paper is reasonably clear, I think the presentation is a bit incomplete and can be made more succinct. For instance, the main algorithm of the paper is not even in the main paper and requires reading the appendix.
2. The algorithm runs in three stages and is quite complicated for solving the tabular MDP problem. Also, I feel uneasy about Stage 1 of the algorithm, since estimating occupancy distributions is a hard problem in practice, eg. the hardness of DICE algorithms in offline RL is precisely due to this.
3. Can you discuss the computationally complexity of each stage, compared to optimal tabular methods like UCBVI? In Line 340, it says the FW subroutine might be expensive, but can you be more concrete and derive a big-Oh running time? My worry is that this paper, while obtaining better instance-dependent statistical bounds sacrifice the practicality of the algorithm. There are also no experiments to validate the practicality of the new coverage assumption.

**Questions:**

1. How is f_{mixed} is defined in Eq. 26 and 27? I couldn't find a clear definition.
2. For Eq. 27, isn't the minimax-optimal result by Azar supposed to be $H^2 SA/\epsilon^2$ trajectories?
3. Where does the current analysis break when extending to more general MDPs, like linear MDPs?

**Limitations:**

See above.

---

> ### Author Rebuttal · Authors · 2023-08-10
>
> Thanks a lot for the helpful comments and valuable feedback.
> Please find below our point-to-point response, which will also be incorporated into the final paper.
>
> 1. In the original submission, the main algorithm was included in the appendix due to the page limit.
> In the final version, we believe that we will have more space, and will definitely include the main algorithm in the main text.
>
> 2. Since we do not need an explicit expression for all occupancy distributions, this stage for estimating $d^{\pi}$ is actually not harder than the remaining stages with respect to both the sample and computational complexities.
> Specifically, for estimating $d^{\pi}$, the sample and computational complexities are $O(K^{\mathsf{on}})$ and $O(K^{\mathsf{on}} + H^7S^8A^3)$, respectively.
> In comparison, the sample and computational complexities of Stage 3 of the offline RL procedure are $O(K)$ and $O(KH)$, respectively.
> Let us explain these complexities of our algorithm in a little more detail below.
>
>  (1). According to Eq. 14, we collect $O(K^{\mathsf{on}})$ trajectories for this procedure, which implies its sample efficiency.
> Note that the current number of trajectories used for this stage is chosen for ease of presentation, and it suffices to use even a even smaller sample size to estimate $d^{\pi}$.
>
>  (2). As for the computational complexity of the occupancy measure in Stage 1, we need to first estimate the transition matrices $P_h$'s with $O(K^{\mathsf{on}})$ complexity. In the following stages, we need to call Algorithm 3 to estimate $\widehat{d}^{\pi}$  for each $\pi$ we encounter. When computing $\pi^{\mathsf{imitate}}$, we need to calculate $\widehat{d}^{\pi}$ for  $T_1=T_{\mathsf{max}}\times T_2$ times, where $T_2$ denotes the number of iterations for calculating $\mu^{t+1}$ in Eq. 20b;
> 		in comparison, the number of estimating $d^{\pi}$ to yield $\pi^{\mathsf{explore}}$ in Eq. 22 is much smaller.
> For each $\widehat{d}^{\pi}$, it needs $O(HS^2A)$ computation.
> As discussed in the following item, with a slight modification on the target Eq. 18, we can make $T_{\mathsf{max}} = \widetilde{O}(H^2S^2)$, and it takes $T_2=O(H^4S^4A^2)$ iterations for calculating $\mu^{t+1}$.
> In summary, the total computational complexity is $O(K^{\mathsf{on}} + H^7S^8A^3)$.
>
>
> 3. With a slight modification on the target Eq. 18 as follows
> \begin{align}
> \mu^{\mathsf{imitate}}\approx\arg\min_{\mu\in\Delta(\Pi)}\sum_{h=1}^{H}\sum_{s\in\mathcal{S}} \max_{a \in \mathcal{A}}\frac{\widehat{d} _ {h}^{\mathsf{off}}(s,a)}{\frac{1}{KH} + O\big(\frac{1}{SH}\big)\widehat{d} _ {h}^{\mathsf{off}}(s,a) + \mathbb{E} _ {\pi^{\prime}\sim\mu^{\mathsf{explore}}}\big[\widehat{d} _ {h}^{\pi^{\prime}}(s,a)\big] + \mathbb{E} _ {\pi^{\prime}\sim\mu}\big[\widehat{d} _ {h}^{\pi^{\prime}}(s,a)\big]},
> \end{align}
> we can find a good enough $\pi^{\mathsf{imitate}}$ with $T_{\mathsf{max}} = \widetilde{O}(H^2S^2)$ for $\eta \asymp \frac{1}{H^2S^2}$ and $O(H^4S^4A^2)$ Frank-Wolfe updates for $\alpha \asymp \frac{1}{H^3S^3A^2}$.
> Then we have the following computational complexity for each stage: $O(K^{\mathsf{on}} + H^7S^8A^3 + K^{\mathsf{off}}H)$ for Stage 1, $O(H^7S^7A^3)$ for Stage 2, and $O(KH)$ for Stage 3.
> Hence, compared to the typical offline RL algorithms like UCB-VI (which has $O(KHS^2A)$ computational complexity(, the computational complexity of our algorithm is better when $K $ becomes sufficiently large.
> We will add a remark to discuss the computational runtime in the final paper.
>
> 4. $f_{\mathsf{mixed}}$ is defined on the left-hand side of Eq. 26 as follows:
> \begin{align}
> f_{\mathsf{mixed}} := \frac{H^3SA\min\\{H\sigma, 1\\}}{\varepsilon^2} + \frac{H^3SC^{\star}(\sigma)}{\varepsilon^2}.
> \end{align}
> We will make this definition more clear in the revised paper.
>
> 5. Regarding the minimax sample complexity limit, we note that: (i) for time-homogeneous MDPs (so that the transition matrix is the same across all steps), the minimax sample complexity is $H^2SA/\varepsilon^2$;
> (ii) for time-inhomogeneous MDPs, the minimax sample complexity should be $H^3SA/\varepsilon^2$, which is the case considered herein.
>
>
> 6. It is possible to extend the idea developed in this work to accommodate linear MDPs.
> There are several important steps needed. Firstly, we need to find an efficient estimation method for the transition kernels of  the linear MDP. Secondly, we need to use the estimated MDP to imitate the behavior policy of the offline dataset and also learn a good exploration policy.
> Finally, we need to invoke (or develop) a sample-optimal offline RL algorithm in the case of linear MDPs to find the optimal policy based on the collected sample trajectories.
> Accomplishing all these details needs substantial efforts, which we leave for future work.

---

> > ### Comment · Area_Chair_tT9k · 2023-08-17
> > **Follow up**
> >
> > Dear Reviewer,
> >
> > We would appreciate if you would you be so kind as to acknowledge and respond to the authors' rebuttal. This is crucial to ensure the reviewing process is conducted adequately.
> >
> > AC

---

> > > ### Comment · Reviewer_Lcd6 · 2023-08-17
> > >
> > > Many thanks for the response, which clarified a few points. My main concern regarding complexity of algorithm (esp. since the paper is just for tabular MDPs) still remains, and I'm on the fence on whether the extra gain (in theory) warrants this complicated approach. I thus keep my score.

---

> > > > ### Author Response · Authors · 2023-08-18
> > > > **Computational complexity of our algorithm**
> > > >
> > > > Dear Reviewer Lcd6,
> > > >
> > > > Thanks for reading our response!  Regarding the computational complexity, we would like to emphasize again that our computational complexity is actually much lower than the popular model-based online RL algorithm UCBVI (more precisely, the asymptotic computational complexity of ours is $KH$, and the computational cost of UCBVI is $KHS^2A$, meaning that our algorithm is actually $S^2A$ times faster than UCBVI)

---

### Decision · Program_Chairs · 2023-09-21

**Decision:**

Accept (poster)

**Comment:**

This work studies the setting of hybrid RL by providing tighter than existing bounds for tabular MDPs in the case that offline data and online interactions are allowed. The bounds are based on a novel and weaker than existing concentrability notion called single-policy partial concentrability that characterizes the statistical complexity of Hybrid RL in this scenario. Although there was no uniform agreement, the reviewers generally believed this to be a worthwhile contribution to appear at Neurips 2023.